# Impact of Snow Thermal Conductivity Schemes on pan-Arctic Permafrost Dynamics in CLM5.0

Adrien Damseaux[1,2,3], Heidrun Matthes[1], Victoria R. Dutch[4,5], Leanne Wake[5], and Nick Rutter[5]

[1]Alfred-Wegener-Insitut (AWI), Potsdam, Germany
[2]Karlsruhe Institute of Technology (KIT), IMK-IFU, Garmisch-Partenkirchen, Germany
[3]Institute of Physics and Astronomy, University of Potsdam, Potsdam, Germany
[4]School of Environmental Sciences, University of East Anglia, Norwich, UK
[5]Department of Geography and Environmental Sciences, Northumbria University, Newcastle, UK

**Correspondence:** Adrien Damseaux (adrien.damseaux@kit.edu)

**Abstract.** The precise magnitude and timing of permafrost-thaw-related emissions and their subsequent impact on the global climate system remain highly uncertain. This uncertainty stems from the complex quantification of the rate and extent of permafrost thaw, which is influenced by factors such as snow cover and other surface properties. Acting as a thermal insulator, snow cover directly influences surface energy fluxes and can significantly impact the permafrost thermal regime. However, current Earth System Models often inadequately represent the nuanced effects of snow cover in permafrost regions, leading to inaccuracies in simulating soil temperatures and permafrost dynamics. Notably, CLM5.0 tends to overestimate snowpack thermal conductivity over permafrost regions, resulting in an underestimation of the snow insulating capacity. By using a snow thermal conductivity scheme better adapted for snowpack typically found in permafrost regions, we seek to resolve thermal insulation underestimation and assess the influence of snow on simulated soil temperatures and permafrost dynamics. Evaluation using two Arctic-wide soil temperature observation datasets reveals that the new snow thermal conductivity scheme reduces the cold soil temperature bias (RMSE = 3.17 to 2.4°C, using remote sensing data; RMSE = 3.9 to 2.19°C, using in-situ data), demonstrates robustness through sensitivity analysis under lower tundra snow densities, and addresses the overestimation of permafrost extent in the default CLM5.0. This improvement highlights the importance of incorporating realistic snow processes in land surface models for enhanced predictions of permafrost dynamics and their response to climate change.

## 1 Introduction

Permafrost contains between 677 and 949 Pg of soil organic carbon (SOC) in the upper few meters, roughly twice as much carbon as the atmosphere (Palmtag et al., 2022). As permafrost thaws with increased temperature, SOC becomes available for microbial decomposition, resulting in the release of large amounts of greenhouse gases into the atmosphere, which, in turn, increase surface temperatures. This permafrost-carbon feedback will likely accelerate climate change, however, the precise magnitude and timing of these emissions and their subsequent impact on the global climate system remain uncertain (Schuur et al., 2015).

A key aspect of this uncertainty is the complex quantification of the rate and extent of permafrost thaw. Predicting how the permafrost thermal regime will respond to ongoing climate change is particularly challenging given its high sensitivity to surface properties (Barrere et al., 2017). Among these, snow cover acts as an important moderator by directly influencing surface energy fluxes between the air and the soil. Functioning as a thermal insulator, snow cover can limit heat loss from the ground during winter (Lawrence and Slater, 2010; Li et al., 2021; Royer et al., 2021), but its insulating properties are highly variable and insufficiently detailed in Earth System Models (Barrere et al., 2017).

The insulating efficiency of snow cover increases with thickness, reaching its peak insulation capacity at around 25 cm of depth (Slater et al., 2017), depending on the [micro]structure and stratigraphy of the snowpack. As denser snow has fewer air voids, resulting in fewer insulating air pockets, thermal conductivity also tends to increase with density (Adams and Sato, 1993). As a result, heat is transferred more efficiently through a dense snow matrix. Snowpack in Arctic tundra environments typically consists of two main parts: depth hoar and wind slab (Sturm et al., 1995; Domine et al., 2018). Depth hoar forms towards the base of the snowpack due to strong vertical temperature gradients and water vapor fluxes. Wind slab forms due to snow compaction from the strong Arctic wind transport and deposition. Depth hoar crystals have large, faceted, and often cup-shaped grains with low density, making them poor heat conductors, while wind slab layers have higher density, resulting in better heat conductivity and decreased insulation properties.

Studies show that state-of-the-art land surface models (LSMs) and snowpack models, including CLM5.0 (Lawrence et al., 2019), Crocus (Vionnet et al., 2012), and SNOWPACK (Bartelt and Lehning, 2002), struggle to represent these two phenomena (Barrere et al., 2017; Gouttevin et al., 2018; Domine et al., 2019; Dutch et al., 2022; Schädel et al., 2024). Notably, vertical density profiles simulated by these models often exhibit significant discrepancies from observed snow density, both in the top wind slab and bottom depth hoar layers of the snowpack (Dutch et al., 2022). Efforts such as those by Brondex et al. (2023) aim to address this issue by developing finite-element models to improve the representation of interactions between heat conduction and water vapor diffusion in snowpack models. However, this extensive work is still in early stages, and neglecting the role of depth hoar in providing thermal insulation properties to Arctic tundra snowpacks can have large consequences for soil temperature representation within LSMs (Gouttevin et al., 2018; Royer et al., 2021; Dutch et al., 2022).

The insulating capacity of a snowpack is determined by the snow thermal conductivity: a critical parameter influencing heat exchange between the soil and atmosphere. Previous studies have highlighted the high sensitivity of LSMs' soil temperature simulations to this parameter (Wang et al., 2013; Paquin and Sushama, 2015), identifying it as a significant source of uncertainty (Langer et al., 2013; Barrere et al., 2017; Domine et al., 2019; Hu et al., 2023). In models, it is expressed as the effective snow thermal conductivity $K_{\text{eff}}$, which aims to account for all heat-transfer processes in a single vertical dimension. Snow exhibits a low $K_{\text{eff}}$, generally falling within the range of 0.01-0.7 $Wm^{-1}K^{-1}$ (Gouttevin et al., 2018); tundra snowpacks typically display $K_{\text{eff}}$ values toward the lower end of this range (Sturm et al., 1997; Domine et al., 2016; Dutch et al., 2022). Numerous studies (Yen, 1981; Jordan, 1991; Sturm et al., 1997; Calonne et al., 2011; Fourteau et al., 2021) describe empirical relationships between $K_{\text{eff}}$ and snow density based on experiments made in laboratories on different snowpacks around the world. Among them, Sturm et al. (1997) derived a regression equation relating density and thermal conductivity based on 488 measurements of pan-Arctic and Antarctic seasonal snow:

$$K_{\text{eff}} = \begin{cases} 0.023 + 2.23 \times 10^{-4} \cdot \rho_{sno}, & \text{if } \rho_{sno} < 156 \\ 0.138 - 1.01 \times 10^{-3} \cdot \rho_{sno} + 3.233 \times 10^{-6} \cdot \rho_{sno}^2, & \text{if } 156 \leq \rho_{sno} \leq 600 \end{cases} \qquad (1)$$

where $\rho_{sno}$ is the snow density in $kg\ m^{-3}$. The Sturm equation stands out due to its notably lower $K_{\text{eff}}$ compared to other relationships based on non-Arctic snowpacks (Fig. A1), particularly within the range of typical Arctic tundra snowpack densities, 150 to 300 $kg\ m^{-3}$.

Barrere et al. (2017) demonstrates that the Sturm et al. (1997) equation better fits their measurements in the Qarlikturvik valley because it is specifically based on tundra snow characteristics. In contrast, equations commonly used by many LSMs (e.g., Anderson (1976) equation in ORCHIDEE (Guimberteau et al., 2018), Mellor (1977) equation in CLASSIC1.0 (Melton et al., 2020), Yen (1981) equation in ISBA (Boone et al., 2016) and JULES (Best et al., 2011), Jordan (1991) equation in CLM5 and ELMv0 (Golaz et al., 2019)), are more adapted to alpine conditions and may not accurately represent pan-Arctic environments. Royer et al. (2021) conducted a sensitivity experiment involving five modified settings in a LSM-Snowpack coupled model, one of which incorporated the Sturm et al. (1997) equation. Their assessment demonstrated only slight improvements in soil temperature; however, it is difficult to isolate the specific impact of the Sturm et al. (1997) equation in their study amongst the other modified parameters. Conversely, Dutch et al. (2022) conducted a comparative analysis of different snow thermal conductivity schemes with CLM5.0 using in-situ measurements from Trail Valley Creek, Northwest Territories, Canada and found that the CLM5.0 default scheme (Jordan, 1991) overestimates snow thermal conductivity by a factor of 3 compared to observations, consequently inducing a cold bias in the wintertime soil temperature simulations. When replacing the default scheme with the formulation proposed by Sturm et al. (1997), significant improvements were observed in wintertime soil temperature simulations. In addition, Paquin and Sushama (2015) and Tao et al. (2024) studied the effects of integrating the Sturm et al. (1997) equation into the LSM CLASS (Verseghy, 1991) and ELM (Golaz et al., 2019), respectively, further underscoring the significant sensitivity of soil temperatures to snow thermal conductivity. Moreover, Paquin and Sushama (2015) demonstrate that the Sturm et al. (1997) scheme effectively mitigates winter soil temperature biases.

Our study aims to extend Dutch et al. (2022)'s assessment to evaluate the applicability of the Sturm et al. (1997) scheme in CLM5.0 across a broader regional climatological context. We hypothesise that a modification to the CLM5.0 snow thermal conductivity scheme will more effectively capture the sensitivity inherent in Arctic tundra snow, thereby restoring a more accurate thermal insulating function of the snowpack and improving soil temperature and permafrost dynamics represented by the model. To realise this endeavor, we present a CLM5.0 sensitivity experiment using the Sturm et al. (1997) snow thermal conductivity scheme and evaluate simulations using Arctic-wide soil temperature in-situ observations and remote sensing data. Additionally, we conduct a sensitivity analysis on snow density to test the robustness of our results for potential lower bulk snow densities characteristic of tundra environments.

## 2 Methods and data

### 2.1 Model description

This study uses the Community Land Model (CLM5.0), which is part of the Terrestrial System Model (CTSM; https://github.com/ESCOMP/CTSM). CLM5.0 is released by the National Center for Atmosphere Research (NCAR) and is the default land component of the community-developed earth system model CESM2. CLM5.0 is a process-based model of the land surface and the terrestrial biosphere that calculates water, energy, and carbon fluxes between the surface and different soil layers. A comprehensive model description and global evaluation can be found in Lawrence et al. (2019) and in the technical description (Lawrence et al., 2018).

#### 2.1.1 Soil

The model soil stratigraphy includes 25 soil layers distributed geometrically, with thinner layers at shallower depths and larger layers at greater depths up to -50 meters. CLM5.0 has an increased soil layer resolution compared to CLM4.5, particularly in the upper -3 meters, to more accurately represent the Active Layer Thickness (ALT) in permafrost areas (Lawrence et al., 2019).

The heat transfer equation (Eq. 6.4 in Lawrence et al., 2018) is numerically solved to compute soil temperatures across the 25-layer column, assuming a heat flux of zero at the bottom of the soil column. Soil temperatures are evaluated at each time step to assess phase changes in water and account for latent heat uptake and release. Hydrological calculations are conducted in the upper 20 soil layers, while the 5 bedrock layers are impermeable to water. Vertical soil moisture transport in the model is driven by the water balance equation of the whole column system, considering infiltration, surface and subsurface runoff, gradient diffusion, gravity, canopy transpiration through root extraction, and interactions with groundwater, respecting the conservation of mass. Vertical soil water flux is computed using Darcy's Law.

The model defines soil thermal and hydraulic conductivities using mineral soil parametrizations dependent on soil texture (sand, clay, and silt fractions) and organic matter density, as derived from Hugelius et al. (2013). These fractions vary across the first 10 layers but remain constant in the subsequent 15 layers.

#### 2.1.2 Snow

The snow module in CLM5.0, described in van Kampenhout et al. (2017) and Lawrence et al. (2019), includes physical processes such as snow accumulation, compaction (due to overburden pressure and drifting snow), refreezing, melting, and sublimation. However, the snow module does not take into account water vapor flux through snow. The CLM5.0 snow module uses a multi-layer approach that discretises the snowpack into a maximum of 12 layers. Fresh snow density is parametrised by combining a temperature term with a linear wind-dependent density term (van Kampenhout et al., 2017). Snow can densify via four distinct processes: compaction by overburden pressure, compaction by drifting snow, destructive metamorphism, or melt

metamorphism. Furthermore, snow thermal conductivity is solely dependent on snow density and calculated following Jordan (1991) scheme by default:

$$K_{\text{eff}} = \lambda_{\text{air}} + \left(7.75 \times 10^{-5} \rho_{\text{sno}} + 1.105 \times 10^{-6} \rho_{\text{sno}}^2\right) \left(\lambda_{\text{ice}} - \lambda_{\text{air}}\right) \tag{2}$$

where $\lambda_{\text{air}}$, $\lambda_{\text{ice}}$ are the thermal conductivity of air $= 0.023 \, Wm^{-1}K^{-1}$, and ice $= 2.29 \, Wm^{-1}K^{-1}$, respectively. Improve-
120 ments to the CLM5.0 snow module have led to increased bulk snow density across most of the Arctic tundra compared to CLM4.5 (Lawrence et al., 2019).

## 2.2 Model set-up and experiments

The version of CLM used throughout this study is ctsm5.1.dev086. The domain for this study is between latitudes 57-90° N and consists of 204086 grid points with a triangular resolution which varies between 116.3 and 179.4 km$^2$, giving a rectangular
resolution of around 12 km$^2$. This is a similar domain to Birch et al. (2020), who used a coarser resolution.

Default CLM5.0 meteorological forcing data (CRU/GSWP3) are replaced by the finer 31 km$^2$ spatial resolution ERA5 forcings from 1980 to 2021 (Hersbach et al., 2020) at an hourly timestep. To our knowledge, this is the second time that CLM5.0 is used with ERA5 forcings, after Cheng et al. (2023). While this increase in resolution should represent a substantial improvement over previous global reanalysis methods used (Albergel et al., 2018), it also introduces additional uncertainty
since the model was not parametrised with these settings as its default configuration. To start the run in an equilibrium state, a spin-up of 30 years using ERA5 reanalysis (looping from 1980 to 1989 three times) was used before running the model from 1980 to 2021 (42 years).

To reduce computation time, this study uses the satellite phenology (SP) set-up, which does not include complex carbon cycle interactions, and deactivates the land-ice and river routing models. In order to prevent unrealistically high values of snow
heights observed in pan-Arctic non-glaciated islands, the snow initialisation protocol was recalibrated with the snow water equivalent (SWE) reverted to its original value of 0.8 m, instead of 10 m as later proposed in van Kampenhout et al. (2017).

We conducted two simulations: (1) the "control run", and (2) the "Sturm run", where the conventional snow thermal conductivity scheme is replaced with the scheme proposed by (Sturm et al., 1997, Eq.1 herein). To assess the sensitivity of model outputs to snow density, additional simulations were performed using both the Sturm and Jordan thermal conductivity schemes,
with adjustment factors of 0.9 and 0.7 applied to the snow density parameterization to better represent the lower bulk snow densities characteristic of tundra environments. In CLM5.0, the snow density is computed as follows:

$$\rho_{sno} = \text{af} \cdot \left(\frac{\omega_{\text{ice}} + \omega_{\text{liq}}}{\text{frac}_{\text{snow}} \cdot d_{\text{z}}}\right) \tag{3}$$

where af is the adjustment factor used in this sensitivity analysis, $\omega_{\text{ice}}$ is the ice lens mass per unit area in kg/m$^2$, $\omega_{\text{liq}}$ is the liquid water mass per unit area in kg/m$^2$, frac$_{\text{snow}}$ is the fractional snow-covered area, and d$_{\text{z}}$ is the snow layer depth in m.

The simulations were conducted exclusively for the 2006–2010 period, selected due to its robust observational data avail-ability, to balance computational efficiency with model reliability. The four additional runs include: (1) Sturm with af = 0.9, (2) Jordan with af = 0.9, (3) Sturm with af = 0.7, and (4) Jordan with af = 0.7. These sensitivity runs were compared to baseline simulations (with af = 1.0) as part of a broader analysis of snow density impacts on model performance.

## 2.3   Data for model evaluation

The Arctic tundra has long been recognised as a difficult region to study due to its inherent remoteness and scarcity of obser-vations (Matthes et al., 2017; Domine et al., 2019; Royer et al., 2021). Accordingly, the lack of information on snow properties in Arctic tundra regions places a major limitation on permafrost and climate modelling (Domine et al., 2016; Gouttevin et al., 2018). To address this challenge, this paper uses two observation datasets as constraints for CLM5.0 outputs: one derived from remote sensing products and the other obtained through in-situ measurements. Both datasets offer complementary perspectives,
enabling a thorough integration and analysis of soil temperature assessment, including (1) temporal scale variations covering seasonal and annual averages, (2) spatial distributions across a wide geographical area, and (3) depth variations throughout the entire soil column.

### 2.3.1   Remote sensing data

We use grid-based products from the European Space Agency (ESA) Climate Change Initiative (CCI) Essential Climate Vari-
160 ables (ECVs) product database from the CCI+ Permfrost project (Obu et al., 2024). ESA-CCI products encompass ECVs with a high spatial resolution of 1 km$^2$ and include Mean Annual Ground Temperature (MAGT) at distinct ground depths of 1, 5, and 10 meters, Permafrost FRaction (PFR) - proportion of an area covered by permafrost within a grid point, and the ALT - the top layer of soil that thaws during the warm season and freezes during the colder months. Product validation is documented in Heim et al. (2021), with further details on the methods available in Obu et al. (2019). The geographical extent of these products
spans the Northern hemisphere above 30°N within an Arctic stereographic circumpolar projection. The temporal coverage for MAGT, ALT, and PFR time series is from 1997 to 2019 at an annual resolution.

    To compare CLM5.0 simulations to ESA-CCI products, we aggregated ESA-CCI products to the domain grid using a con-servative second-order regridding equation described in Jones (1999). Following the Osterkamp and Romanovsky (1999) def-inition of permafrost as ground that remains at or below 0°C for at least two consecutive years, the presence or absence of
170 permafrost (PFR) at each grid point within CLM5.0 is determined by:

$$\text{PFR} = \begin{cases} 1, & \frac{1}{M}\sum_{y=1}^{M} \min_{z=1,N} \max_{t(y)=1,2Y} T_i(z, t(y)) < 273.15\text{K} \\ 0, & \frac{1}{M}\sum_{y=1}^{M} \min_{z=1,N} \max_{t(y)=1,2Y} T_i(z, t(y)) \geq 273.15\text{K} \end{cases} \tag{4}$$

where $M$ is the number years covered by ESA-CCI product (1997-2019), $z$ is the index for the soil depth, $N$ is the number of depths, $t$ is the index for the days in the year $y$ and the next year $(y+1)$, $Y$ is the number of days in a year, and $T_i(z,(t(y))$ is the temperature depending on the day, depth and grid cell. We first calculated the maximum temperature over a two-year period

for each grid cell and each layer. Then, we calculated the vertical soil temperature minimum to see if there is one continually frozen layer over these two years. From this, we obtained a temperature data grid for each year, which we then averaged over the period spanning 1997 to 2019 to match the duration of ESA-CCI products period. Subsequently, we classified grid points into two categories: those with temperatures below 0°C were designated as permafrost, while those with temperatures above 0°C were classified as non-permafrost. It is worth noting that this method provides a binary definition of permafrost, in contrast to ESA-CCI classification, which offers a quantitative representation of permafrost ranging from 0 to 100% resulting from their ensemble-members experiments. To reconcile this difference, we adopted three permafrost classes for the ESA-CCI data: continuous if greater than 90%, discontinuous if between 50% and 90%, and permafrost free if less than 50%.

To calculate ALT at each grid point within CLM5.0 for each year, a grid of maximum annual soil temperature is computed to identify the first thawed layer (above 0°C) from the basal layer. Subsequently, a spline curve is calculated using the layers above and below the first thawed layer to estimate the actual depth of transition between frozen and thawed soil layers. The resulting ALT for both CLM5.0 and ESA-CCI were then averaged between 1997 to 2019.

To obtain the maps presented in the results section, we subtracted the ESA-CCI grid data from the CLM5.0 simulations for MAGT, PFR and ALT period-averaged products. In addition, we calculated the Mean Absolute Deviation (MAD) and Root Mean Square Error (RMSE) for MAGT and ALT, where predicted values are results from the model and observed values are ESA-CCI products.

### 2.3.2 In-situ soil temperatures

We expanded upon the dataset used by Matthes et al. (2017) using data from the Permafrost Laboratory website (https://permafrost.gi.alaska.edu), the GTN-P database (http://gtnpdatabase.org), the Nordicana D website (https://nordicana.cen.ulaval.ca/), and the Roshydromet network (http://aisori-m.meteo.ru/). The resulting database denoted herein as "295GT" comprised monthly average temperatures for 295 borehole stations over 42 years, from 1980 to 2021, across the entire Arctic (Fig. 1). Soil temperatures have been recorded across 278 distinct depth levels, ranging from -0.01 m to -60 m. When comparing the model results with the 295GT dataset, each station is matched with the nearest grid point, and a linear interpolation is performed using the two closest CLM5.0 depth level options.

## 3 Results

### 3.1 Snow insulation

The winter offset, as defined by Burke et al. (2020), quantifies the difference between the mean soil temperature at 0.2 meters and the mean air temperature during the December to February period. This metric provides valuable insight into the snow insulation capacity and the transfer of heat from the air to the soil during the winter season as represented by a LSM.

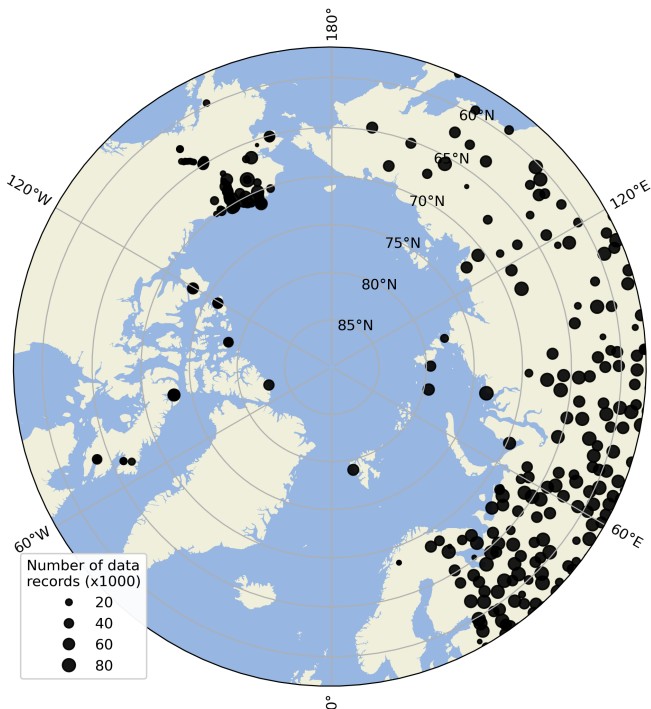

**Figure 1.** Location of the 295 borehole stations used. The size of each point represents the number of data records per station over the whole period and for all depths. The data sets are sourced from the Permafrost Laboratory website, the GTN-P database, Nordicana D, and the Roshydromet network.

The Sturm run demonstrates substantially higher snow insulation across most of the domain, notably in tundra regions, when compared to the control run (Fig. 2). Offset values range between 20 to 35°C over Siberia and 15 to 25°C over Canada and Alaska for the Sturm run, compared to 10 to 20°C over most regions for the control run.

Following the methodology outlined by Wang et al. (2016), Figure 3 illustrates the snow insulation effect between the control and Sturm runs across the 295GT Russian site locations (n = 178), with colours representing various temperature regimes. The disparity in results between the runs is most notable in the cold temperature regime (tundra regions), where the winter offset linearly increases up to 40 cm snow depth and stabilises thereafter in the Sturm run. Conversely, the relationship between snow depth and winter offset is close to linear across all snow depths in the control run.

## 3.2 Soil temperature

Our initial hypothesis suggests that the cold bias in the control run is caused by the Jordan scheme's limitations in associating snow density with thermal conductivity under Arctic conditions, leading to higher-than-expected thermal conductivities which result in lower ground temperatures. To rectify this cold bias, we replaced the Jordan scheme with the Sturm scheme in the Sturm run, aiming to test whether this adjustment can improve the model's representation of ground temperature.

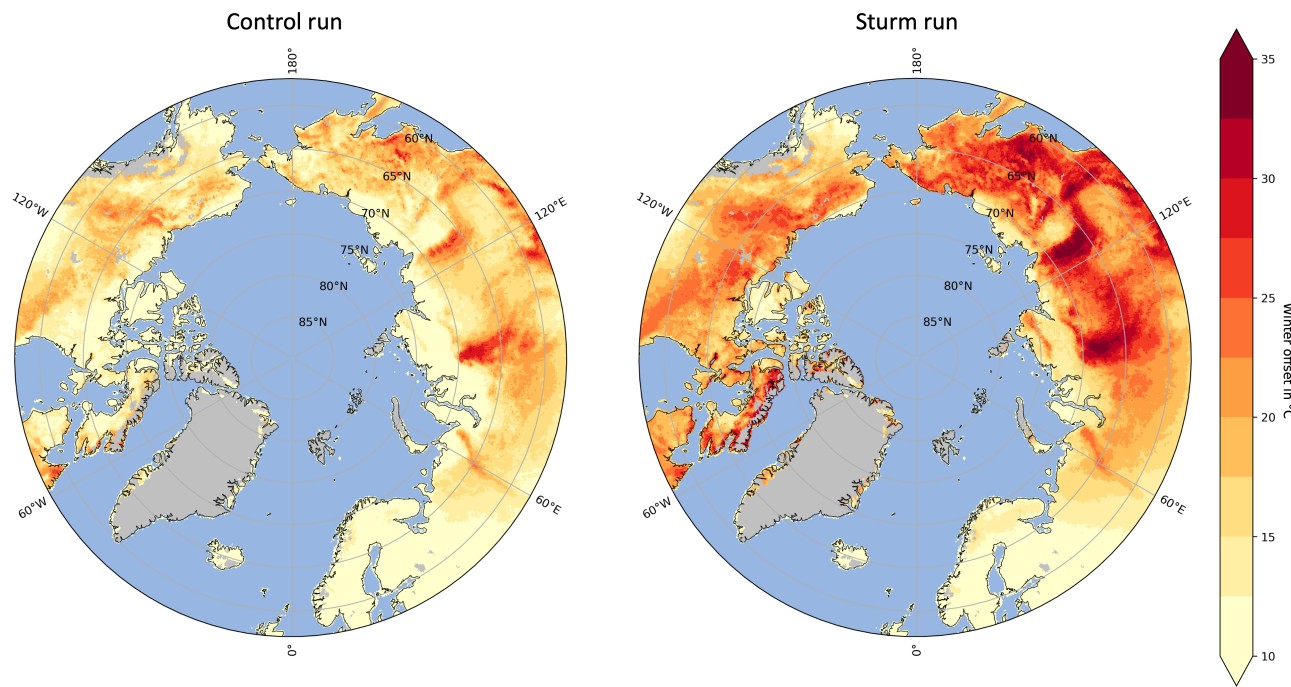

**Figure 2.** Period (1980 to 2021) winter offset for the control run (left) and Sturm run (right), following Burke et al. (2020)'s methodology.

### 3.2.1 Comparison between the Sturm and control runs

During DJF, a significant temperature increase is observed in the Sturm run when compared to the control run (Fig. 4). In the Siberian permafrost region, temperatures increase by 4 to 10°C, while in northern Canada and Alaska, they rise by up to 5°C. In MAM, there is an increase of up to 3°C found mostly over high-altitude areas across the whole domain and on the southwestern Hudson Bay. In JJA and SON, the increase in temperature is much less marked over the whole domain with an increase of temperature from 1 to 2°C, except for mountain areas and western Hudson Bay. In general, we observed a substantial increase in soil temperature in DJF and MAM when snow cover is important. This outcome aligns with our hypothesis that the increased snow insulation in the Sturm run would result in higher DJF soil temperatures.

### 3.2.2 Comparison between the Sturm run and ESA-CCI

The evaluation of the -1 m year-averaged soil temperature (Fig. 5) compares results from the control and Sturm runs against the ESA-CCI dataset. The Sturm run significantly reduces the cold bias observed in the control run within tundra regions, including the West Siberian Plain, Central Siberian Plateau, Yakutsk Basin, Kolyma Lowland, and northern Canada. Similar

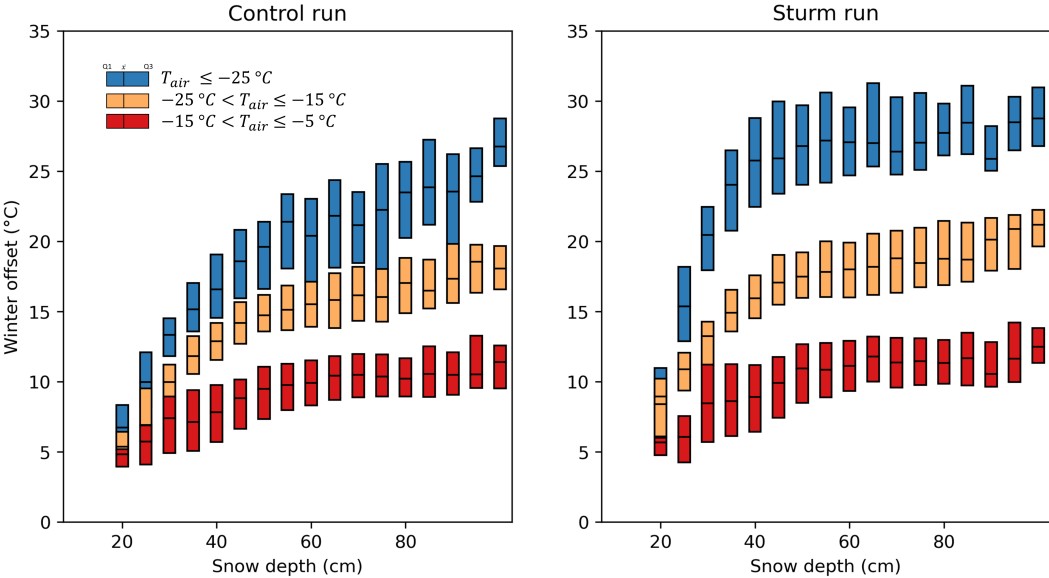

**Figure 3.** Variation between the winter offset with snow depth for the control (left) and Sturm (right) run calculated from the 295GT Russian site locations (n = 178) and 41 individual winters (1981–2021), following a methodology similar to the model comparison undertaken by Wang et al. (2016). Each box plot represents 5 cm snow depth bins and colours indicate different air temperature regimes.

improvements were observed at soil depths of -5 m and -10 m (not shown here). Most regions only have a small cold bias of

up to 2°C.

The MAD and the spread of temperature (RMSE) show a noteworthy improvement, decreasing from 2.63°C in the control run to 1.73°C in the Sturm run for MAD, and from 3.17°C to 2.4°C for RMSE, respectively. However, the RMSE values still remain high. This is probably linked to the pronounced warm bias observed over high-altitude areas (e.g., the Central Siberian Plateau, the Verkhoyansk Range, most of Eastern Siberia, the northern regions of Baffin Island, and the Brooks Range) which

was present in the control run but greatly amplified in the Sturm simulation.

### 3.2.3    Comparison between the Sturm run and the 295GT dataset

In general, the control run reasonably captures the attenuation and delay of the seasonal cycle in soil temperature for period-averaged monthly soil temperatures (Fig. 6) at various depth levels (-20 cm, -80 cm, -160 cm, and -320 cm). However, it consistently exhibits a cold bias of a similar amplitude across all seasons and depths (MAD = 3.23°C, RMSE = 3.32°C for

240    -20 cm; MAD = 4.35°C, RMSE = 4.35°C for -320 cm). The Sturm run effectively minimised the bias gap introduced by the control run, particularly during DJF and within the uppermost soil layers (MAD = 1.76°C, RMSE = 1.93°C for -20 cm). Once the snow has melted out in JJA, the impact of our experiment on snow thermal conductivity decreases, as expected. The slight bias reduction that persists after snowmelt can be attributed to soil temperature memory. In addition, the improvement is less

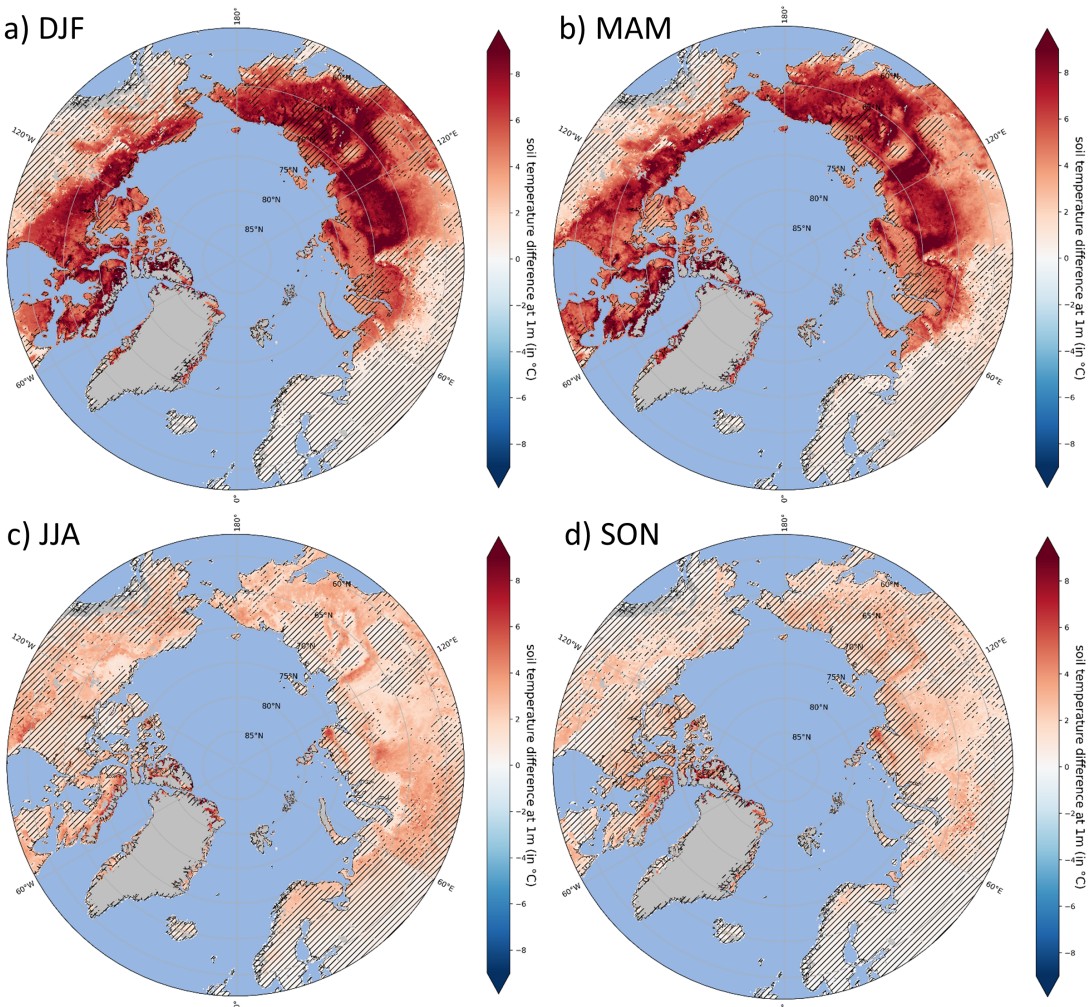

**Figure 4.** Period averaged (1980-2021) soil temperature difference between the Sturm and control runs at -1 m depth for four seasons: a) December, January, February (DJF), b) March, April, May (MAM), c) June, July, August (JJA) and d) September, October, November (SON). Darker red indicates that the Sturm run is warmer than the control run. The grey mask represents glaciers. Hatched areas represent non-significant results compared to the time series (p-values < 0.95).

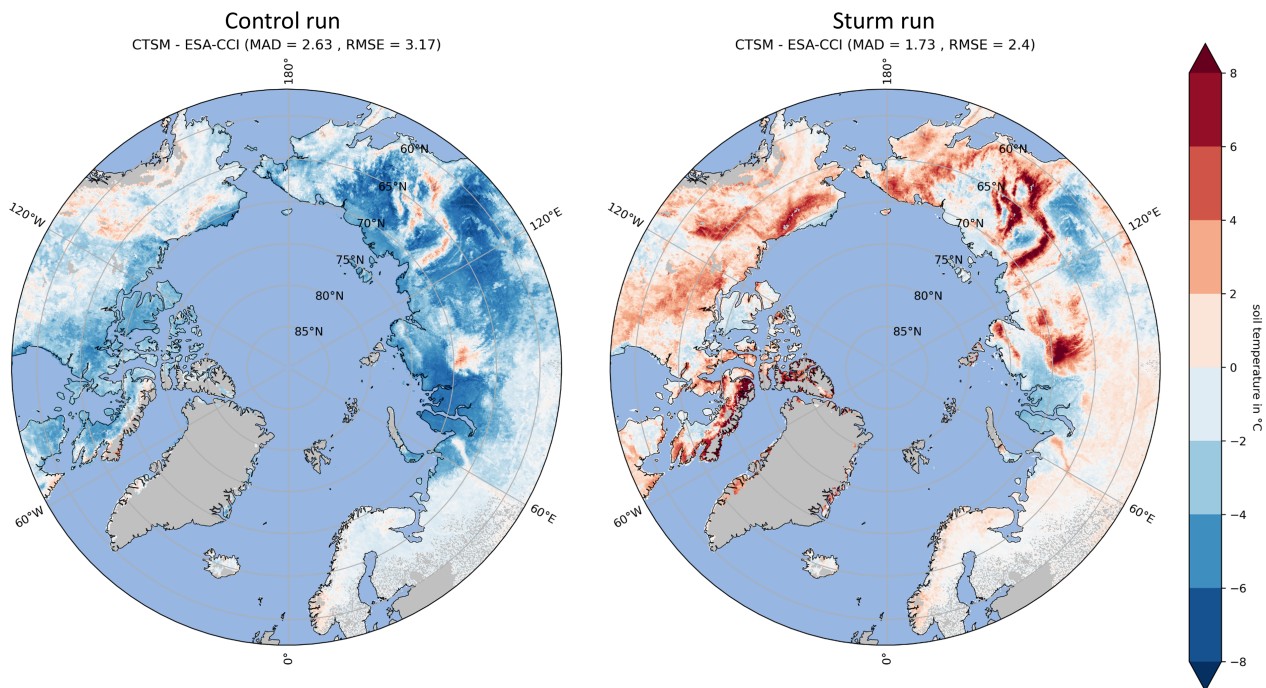

**Figure 5.** Period (1997 to 2019) MAGT at -1 m depth, with the difference between CTSM and ESA-CCI in °C for the control run (left) and Sturm run (right). Darker blue indicates that CTSM soil temperature is colder than ESA-CCI. ESA-CCI data are aggregated on the CTSM grid using a conservative second-order regridding method.

pronounced in deeper layers (MAD = 2.55°C, RMSE = 2.57°C for -320 cm), as the properties of soil increasingly dominate
snow insulation properties at depth. Furthermore, there is a notable positive bias of up to 2°C observed in the top -20 cm soil layer during DJF. On average, the RMSE across the four soil layers decreases from 3.9°C in the control run to 2.19°C in the Sturm run.

### 3.3   Sensitivity analysis to snow density

The sensitivity analysis to snow density shows that the Sturm parameterisation regularly yields lower RMSE values compared
to Jordan (blue cells in Fig. 7). This improvement is most pronounced during winter months (FMA) in deeper layers of soil.
As snow density is reduced, the relative benefit of Sturm over Jordan diminishes, particularly in JFMA months at soil depths of
-20 cm and -80 cm. However, the Sturm parameterisation leads to a lower soil temperature error for most months and depths.
During summer months (without snow cover), the winter influence of the Sturm parametrization continues, simulating a lower
temperature error than that of Jordan, particularly in deeper soil layers.

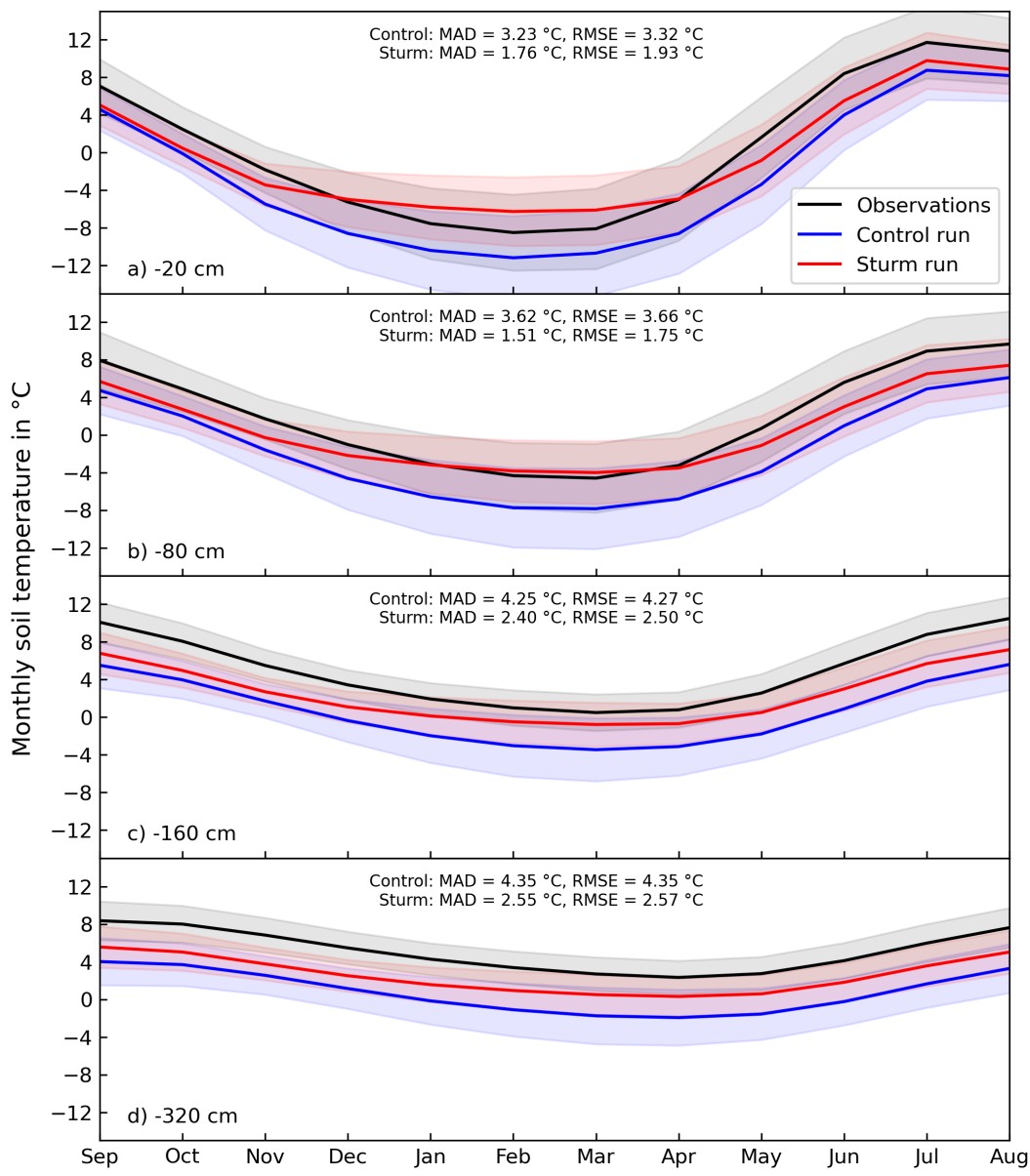

**Figure 6.** Period averaged (1980-2021) of monthly soil temperature for the observations (black), control run (blue) and Sturm run (red) at 4 different depths: a) -20 cm, b) -80 cm, c) -160 cm and d) -320 cm. Each of these represents an average of depth ranges as follow: -20 cm = [0 - 40 cm], -80 cm = [41 - 120 cm], -160 cm = [121 - 200 cm], and -320 cm = [201 - 440 cm]. The shaded area represents the standard deviation over all years. All values and skill scores (MAD, RMSE) come from an average of the 295 stations through the full period.

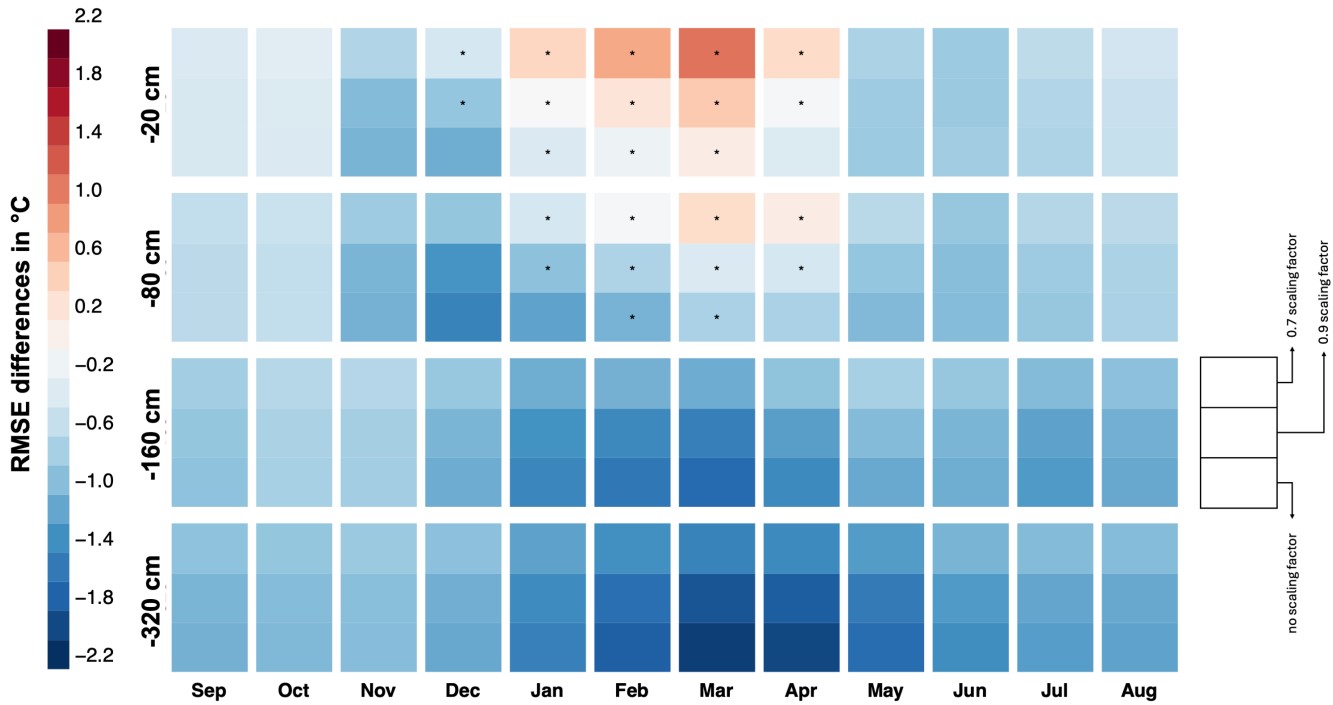

**Figure 7.** Period-averaged (2006–2010) differences in monthly soil temperature RMSE (Sturm minus Jordan) across 295 stations. Each row represents a different depth (at -20, -80, -160, and -320 cm), while each column represents a different month average. Each cell represents a different adjustment factor: 0.7 (top), 0.9 (middle), and no adjustment factor - default (bottom). Cells with positive MAD values in the Sturm run (overshoots) are marked with an asterisk (*). Darker blue indicates improved RMSE scores in Sturm relative to Jordan.

### 3.4 Permafrost extent

There is strong agreement between the control run and ESA-CCI permafrost extents, with 93% of the two datasets overlapping, including the discontinuous Arctic permafrost regions (Fig. 8). However, the control run slightly overestimates permafrost extent in the southern regions of Alaska, Canada, and particularly Siberia.

For the Sturm run, the overestimation of permafrost made by the control run has been resolved to the detriment of mountainous regions (in red) that have been reclassified as non-permafrost (Fig. 8). In addition, the Sturm run shows a marked loss of discontinuous permafrost (in orange). In total, the Sturm run simulates a permafrost extent area equal to $9.489 \times 10^6$ km$^2$, a strong decrease compared to the control run ($13.358 \times 10^6$ km$^2$) and ESA-CCI ($12.544 \times 10^6$ km$^2$) values.

To supplement our analysis with ESA-CCI permafrost extent products, we compare the results for the control and Sturm runs to the International Permafrost Association (IPA) map (Brown et al., 2002) in Figure A2.

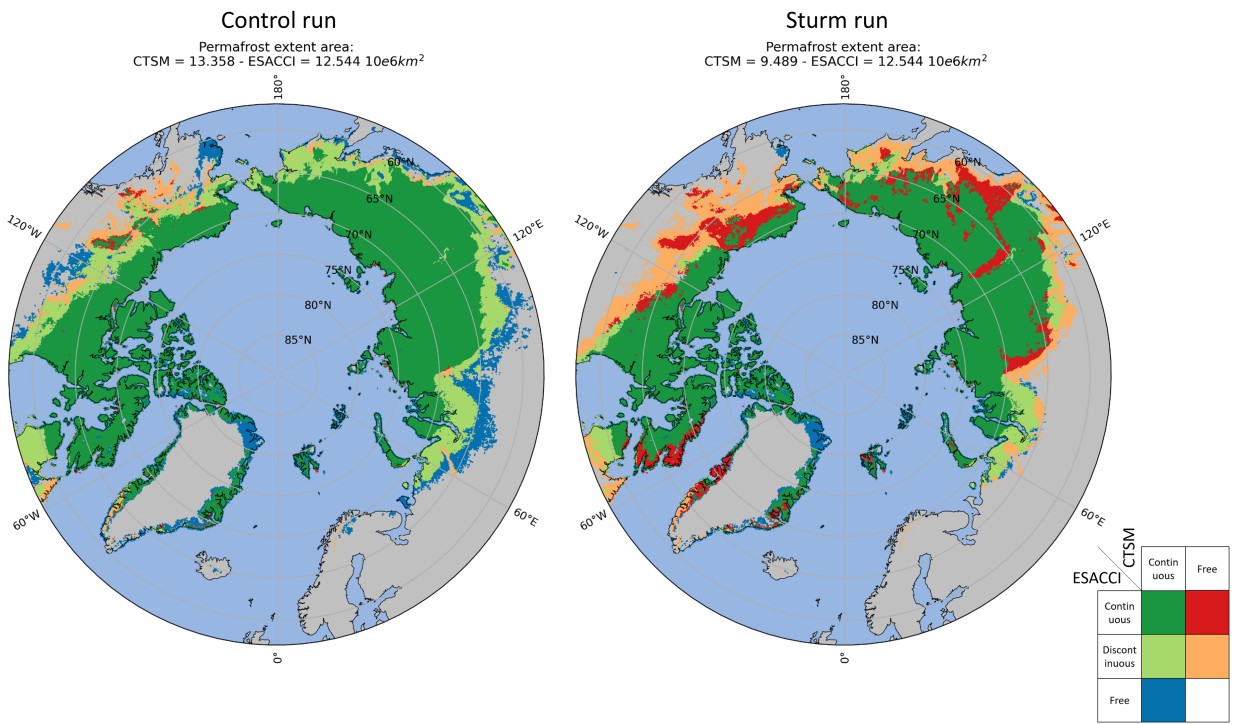

**Figure 8.** Permafrost extent area mask difference between CTSM and ESA-CCI for the control run (left) and Sturm run (right). ESA-CCI data are aggregated on the CTSM grid using a conservative second-order regridding method.

## 3.5 Active Layer Thickness (ALT)

Differences between CLM5.0 and ESA-CCI ALT products indicate a noticeable positive bias increase (Fig. 9) that varies across regions. While minor biases are observed over tundra areas, biases are significantly amplified over mountainous regions and in the southern regions with deep active layers in Siberia. MAD and RMSE scores increase from 0.5 to 1.32 m and from 0.82 to 2.13 m, respectively. Note that we calculated these statistics only within regions identified as permafrost in the Sturm simulation to ensure a direct comparison of identical areas. This approach means that we excluded large regions classified as non-permafrost in the Sturm run from our analysis.

## 4 Discussion

### 4.1 Snow insulation

Earlier findings (Wang et al., 2016; Slater et al., 2017) show that there is a logarithmic relationship between the winter offset and snow depth, reaching an asymptote at a snow depth of approximately 25 cm, according to in-situ observations. In Figure 3,

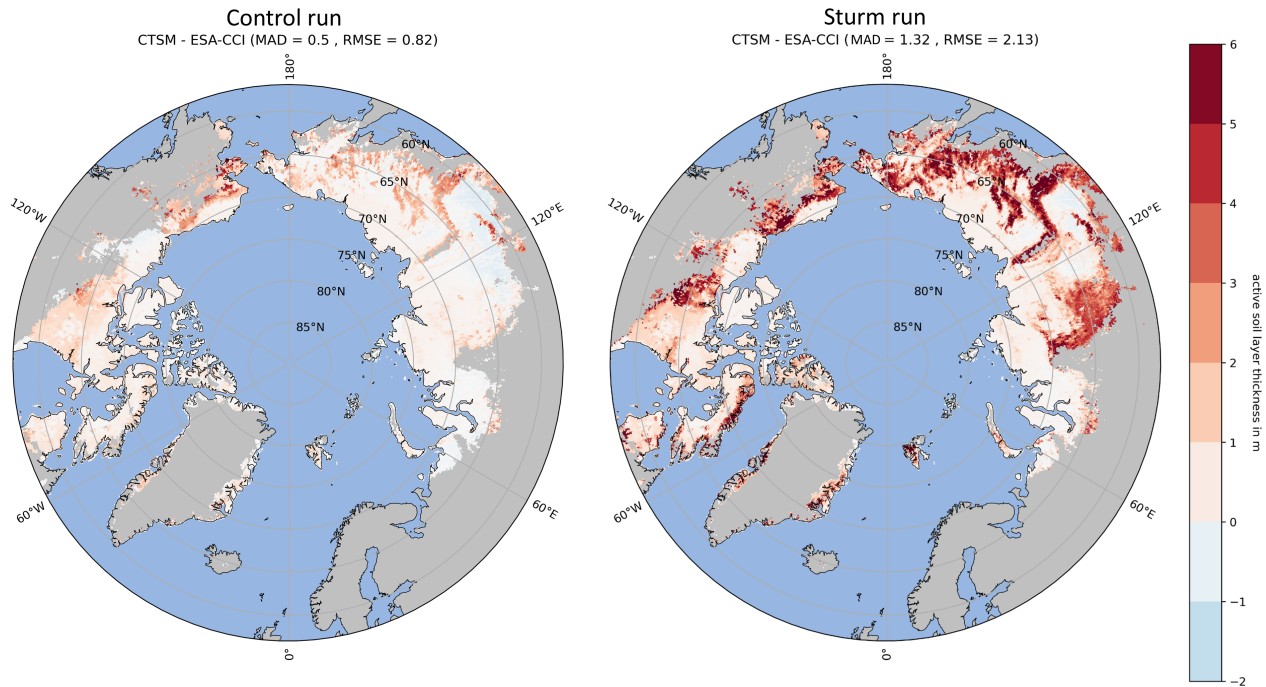

**Figure 9.** Active layer thickness difference between CTSM and ESA-CCI in meters for the control run (left) and Sturm run (right). Darker red indicates that CTSM ALT is deeper than ESA-CCI. ESA-CCI data are aggregated on the CTSM grid using a conservative second-order regridding method. Only regions considered as permafrost in the Sturm simulation are shown to facilitate comparison between the two simulations.

only the Sturm run accurately represents this logarithmic relationship in cold temperature regimes. The control run exhibits a trend closer to a linear relationship, often resulting in an underestimation of snow insulation, which is consistent with findings from other modelling groups (Wang et al., 2016; Slater et al., 2017; Guimberteau et al., 2018; Burke et al., 2020; Pongracz et al., 2021). Interestingly, CESM (using CLM5.0) shows a degradation in the representation of that relationship compared to its previous version using CLM4.5 (Burke et al., 2020). We hypothesise that the underestimation of snowpack density by CLM4.5 (Lawrence et al., 2019) combined with the high thermal conductivity scheme from Jordan (1991) artificially resulted in an adequate snow insulation represented by the model over Arctic tundra regions. The introduction of the new fresh snow density function by van Kampenhout et al. (2017) in CLM5.0 may have unintended consequences, making the bulk snow density too high in Arctic tundra regions, where specific tundra snowpack features like depth hoar are not represented by the model (van Kampenhout et al., 2017). As the snow thermal conductivity scheme remained unchanged from CLM4.5 to CLM5.0, higher snow densities mean that heat energy from the soil can be lost to the atmosphere more efficiently, which may explain the notable cold bias observed in CLM5.0.

The spatial distribution of the winter offset in the Sturm run better aligns with previous findings (Wang et al., 2016) compared to the control run, despite the minimal difference in effective snow depth between the two runs (below +/- 5% in most regions, see Fig. A3). This supports our hypothesis that the snow insulation in the Sturm simulation is considerably increased and generally more representative of tundra snowpacks.

## 4.2 Soil temperature

The magnitude of cold bias observed in the control run is similar to what other modelling groups have shown (Dankers et al., 2011; Burke et al., 2013; Wang et al., 2013; Ekici et al., 2014; Barrere et al., 2017; Guimberteau et al., 2018; Pongracz et al., 2021), especially over colder regions, and tends to be more pronounced in deeper layers. On the other hand, some evaluations of LSMs have reported the absence of such bias (Chadburn et al., 2015; Decharme et al., 2016; Chadburn et al., 2017). However, these studies rely on sparse in-situ measurements (often with an absence of observations in high-latitude regions) that may not fully represent the entire pan-Arctic domain. Other studies evaluating coupled LSM-Snowpack models have shown very good performance in soil temperature representation in the pan-Arctic (Barrere et al., 2017; Royer et al., 2021), underscoring the importance of accurate snow physics, albeit at a higher computational cost. Our results reveal a bias amplitude consistent across all seasons and depths, reflecting findings from prior research (Burke et al., 2013; Paquin and Sushama, 2015). This contrasts with several model studies (Dankers et al., 2011; Wang et al., 2013; Barrere et al., 2017; Guimberteau et al., 2018; Oogathoo et al., 2022), which show larger biases in winter compared to summer. Interestingly, our findings align with similar trends observed in the study by Herrington et al. (2024), which examined the performance of reanalysis soil temperature data across the pan-Arctic and noted a prevalent cold bias.

The results of the Sturm run are consistent with a comparable experiment on snow thermal conductivity conducted by Paquin and Sushama (2015), showing a decrease in wintertime soil temperature bias and a diminishing improvement with depth. However, our results show closer alignment with the observations. Conversely, the model study by Oogathoo et al. (2022) using the Sturm et al. (1997) equation, indicates an underestimation of soil temperature in winter, although their model uses a basic snowpack model with a single layer.

The persistent cold bias in simulated soil temperature in deeper layers may be attributed to several missing snow processes, including more realistic snow metamorphism (Decharme et al., 2016), or upward water vapor mass transfers within the snowpack (Domine et al., 2019). Recent studies have explored these missing processes (Brondex et al., 2023; Fourteau et al., 2024). Additionally, soil processes such as the inclusion of excess ground ice (Lee et al., 2014; Burke et al., 2020), an improved phase-change scheme (Yang et al., 2018; Tao et al., 2021), and the development of adapted-frozen soil thermal conductivity models (He et al., 2021) offer greater potential to improve the soil temperature accuracy in summer and at depth.

In general, the model skill scores perform better against grid-based observations datasets rather than in-situ observations (RMSE = 3.17-3.24°C against ESA-CCI, RMSE = 3.32-4.35°C against 295GT for the control run). The divergence between model outputs and in-situ observations is often attributed to the inherent scale differences. While the model operates at a coarse resolution (12 km$^2$), observations are site-specific. Comparing point observations to model grid points covering a wide area can lead to inaccuracies because individual observations may not fully represent the characteristics of the model grid point

covered area (Dankers et al., 2011; Park et al., 2015). Scale disparities commonly stem from variations in elevation, climate, soil composition, and landscape characteristics, resulting in considerable diversity in soil thermal and hydraulic properties and, consequently, in soil temperature patterns.

Large positive soil temperature biases up to 8°C are particularly noticeable over high-altitude regions in our ESA-CCI evaluation. This discrepancy arises in part from variations in atmospheric forcing resolution between CLM5.0 (12 km$^2$) and ESA-CCI (1 km$^2$); lower resolution models smooth out complex mountain terrain features into larger grid cells, leading to an inadequate representation of temperature in mountain environments (El-Samra et al., 2018). Secondly, the parametrization of the Sturm scheme assumes the presence of basal depth hoar and overlying wind slab, potentially leading to inaccurate

representation of the thermal conductivity of basal and mid-depth snow types typically found in mountainous regions (Sturm et al., 1997). The application of different empirical snow thermal conductivity schemes based on snow types (e.g., tundra or alpine) may address this challenge. However, identifying both meteorological and land surface conditions needed for accurate application of such schemes in a global model like CLM would be challenging.

## 4.3   Sensitivity analysis to snow density

As previously stated, studies show that state-of-the-art LSMs and snowpack models, including CLM5.0, have vertical density profiles often exhibiting significant discrepancies from observed snow density, both in the top wind slab and bottom depth hoar layers of the snowpack. Such discrepancies lead to over-densification in the simulated tundra snowpack. The misrepresentation arises because the scheme does not account for temperature-gradient metamorphism, a process that creates low-density depth hoar layers in tundra snowpacks (Dutch et al., 2022). Without this mechanism, the simulated snow can only increase in density

with age, leading to bulk densities that exceed observed values in these regions. Incorporating temperature-gradient metamorphism in future model developments would likely result in lower simulated snow densities, improving agreement with field observations (Brondex et al., 2023).

    Our sensitivity analysis shows that the RMSE reductions achieved by the Sturm parametrization remain robust, even if future improvements are made to tundra snow densification processes that result in lower bulk densities. This improvement is most

345 pronounced in deeper layers during winter months (FMA), when the cold wave penetrates deeply, emphasizing the relevance for permafrost modelling. This suggests that the improved performance of Sturm over Jordan does not rely on unrealistically high bulk snow density values. However, the increase in RMSE caused by the overestimation of soil temperatures in upper layers during winter months is amplified when snow density is reduced. While this highlights a limitation of the Sturm scheme in certain scenarios, the overall benefits for permafrost modelling outweigh this drawback, particularly in the context of deeper

soil layers where winter thermal dynamics are critical.

## 4.4   Permafrost extent

The comparison between the ESA-CCI permafrost data and our model results involves inherent uncertainties due to differences in spatial resolution. Our land model's grid cells are approximately 100 times larger than those of the ESA-CCI product,

leading to blurred boundaries when aggregating the data. Although the ESA-CCI data itself has uncertainties, with most grid cells having uncertainties below 50%, these are unlikely to outweigh the uncertainties introduced by the resolution mismatch.

Several other modelling groups observe an overestimation of the permafrost extent similar to the control run, as indicated by the CMIP6 intercomparison project on permafrost physics (Burke et al., 2020), although not all models show this behavior. While the Sturm run provides some mitigation of this pattern, some continuous and discontinuous permafrost areas over mountains and southern Alaska, Canada, and Siberia are lost. The issue may arise from the presence of warm permafrost in the southern edge where ground temperatures approach 0°C and the soil moisture content is high. Over those regions, the accuracy of ESA-CCI products is affected because latent heat effects slow down potential thaw, which increases the disequilibrium between atmospheric and ground temperatures (Obu et al., 2019). The area simulated in this study is similar that modeled by Paquin and Sushama (2015) in their Sturm experiment; however, their high-altitude regions remain classified as permafrost.

## 4.5 Active layer

In general, both CLM5.0 configurations show a tendency to overestimate maximum thaw depth, a trend exacerbated in the Sturm run in high-altitude and southern regions. This discrepancy has been observed in many other LSM studies (Dankers et al., 2011; Ekici et al., 2014; Chadburn et al., 2015; Paquin and Sushama, 2015; Guimberteau et al., 2018; Burke et al., 2020; Tao et al., 2024). Using a knowledge-based hierarchical optimisation strategy on a series of parameters (precipitation-phase partitioning, snow compaction, and snow thermal conductivity) and input data (climate forcings and SOC density profile), Tao et al. (2024) effectively enhances ALT results across more than 100 pan-Arctic sites in their LSM. While their methodology shows promise, its implementation across various model setups and models will require thoughtful adaptation and adjustments.

CLM5.0 performs better in high-latitude tundra regions compared to other modelling groups, which often display more pronounced regional biases. Notably, our study is the first to evaluate a LSM's ALT against a grid-based observation product, whereas most other studies to date compare their ALT results to in-situ stations, e.g. CALM in Shiklomanov et al. (2012). The discrepancy observed in southern regions may also be attributed to challenges faced by ESA-CCI data methods, like probing and ground-penetrating radar, in accurately measuring ALT in regions with deeper active layers (Liu et al., 2024). Our findings highlight the critical need for diverse, regionally tailored observational datasets to refine model performance and better capture the complexities of permafrost dynamics.

## 5 Conclusions

With the growing need to assess the substantial impact of permafrost-carbon feedbacks on global climate, it is increasingly important for land surface models (LSMs) to accurately represent ground temperature in permafrost tundra regions. Snow plays a critical role over these regions, providing thermal insulation during winter, which has substantial implications for heat exchange between the atmosphere and the soil. However, Earth System Models (ESMs) often lack sufficient detail regarding the spatial and temporal variability of snow insulation, among other factors.

Building upon a site experiment at Trail Valley Creek (Dutch et al., 2022), this paper applies the Sturm et al. (1997) relationship between snow thermal conductivity and density to the entire pan-Arctic domain, as it is better suited to the snow density profile found over Arctic tundra permafrost regions. Our aim was to study the impact of this scheme on simulated soil temperatures and permafrost dynamics, thereby improving the model's performance in reproducing snow physics over Arctic tundra regions.

The integration of the Sturm et al. (1997) snow thermal conductivity scheme within CLM5.0 resulted in a reduction of cold biases and a closer alignment of model outputs with observational datasets (against remote sensing data, RMSE decreases from 3.17 to 2.4°C; against in-situ data, RMSE decreases from 3.9 to 2.19°C). Our sensitivity analysis to snow density further validates the robustness of the Sturm parameterization, demonstrating that its improvements persist even when accounting for potentially lower bulk snow densities in tundra environments. Furthermore, the Sturm experiment effectively addresses the overestimation of permafrost observed in the control run in southern Siberia and Canada. However, large areas over discontinuous permafrost and mountainous regions were reclassified as non-permafrost. Altogether, the Sturm run simulates a permafrost extent area of $9.489 \times 10^6$ km$^2$, a significant decrease compared to both the control run ($13.358 \times 10^6$ km$^2$) and ESA-CCI ($12.544 \times 10^6$ km$^2$) values. In addition, we observed a notable increase in ALT bias, primarily in mountainous areas. We attribute the bias observed over high-altitude regions to two possible factors: (1) differences in the resolution of the atmospheric forcing data used between ESA-CCI and CLM5.0 and (2) the newly implemented snow scheme may not be ideally suited for mountainous regions.

While the Sturm parametrization offers a substantial improvement in addressing cold biases and enhancing the simulation of snow insulation in Arctic regions, it is not a panacea. Future advancements in the CLM snow scheme, particularly in the representation of snow stratigraphy and processes such as water vapor transport, will be necessary to further refine these simulations and improve model accuracy. The value of improved tundra snow thermal representation in a LSM needs testing within a fully-coupled ESM to understand how consequent changes in simulated soil temperatures impact vegetation (Jin et al., 2021), river flows (Rawlins and Karmalkar, 2024), permafrost-thaw-related $CO_2$ emissions (Dutch et al., 2023), and consequently, climate feedbacks (Schädel et al., 2024). Overall, our findings underscore the importance of refining snow-related processes in LSMs to enhance broader understanding of permafrost dynamics in the context of climate change.

*Code availability.* The model version used in this study is available at https://github.com/AdrienDams/CTSM/tree/levante. The algorithms used to compare the observation datasets with our model results can be found at https://github.com/AdrienDams/cegio/tree/sturm-paper.

*Data availability.* Post-processed model simulations and observations products from ESA-CCI, as well as our 295GT dataset, are available at https://www.wdc-climate.de/ui/entry?acronym=DKRZ_LTA_049_dsg0001. Additional simulations for the sensitivity analysis to snow density are available on request.

## 415  Appendix A: Additional figures

### A1  Snow thermal conductivity schemes

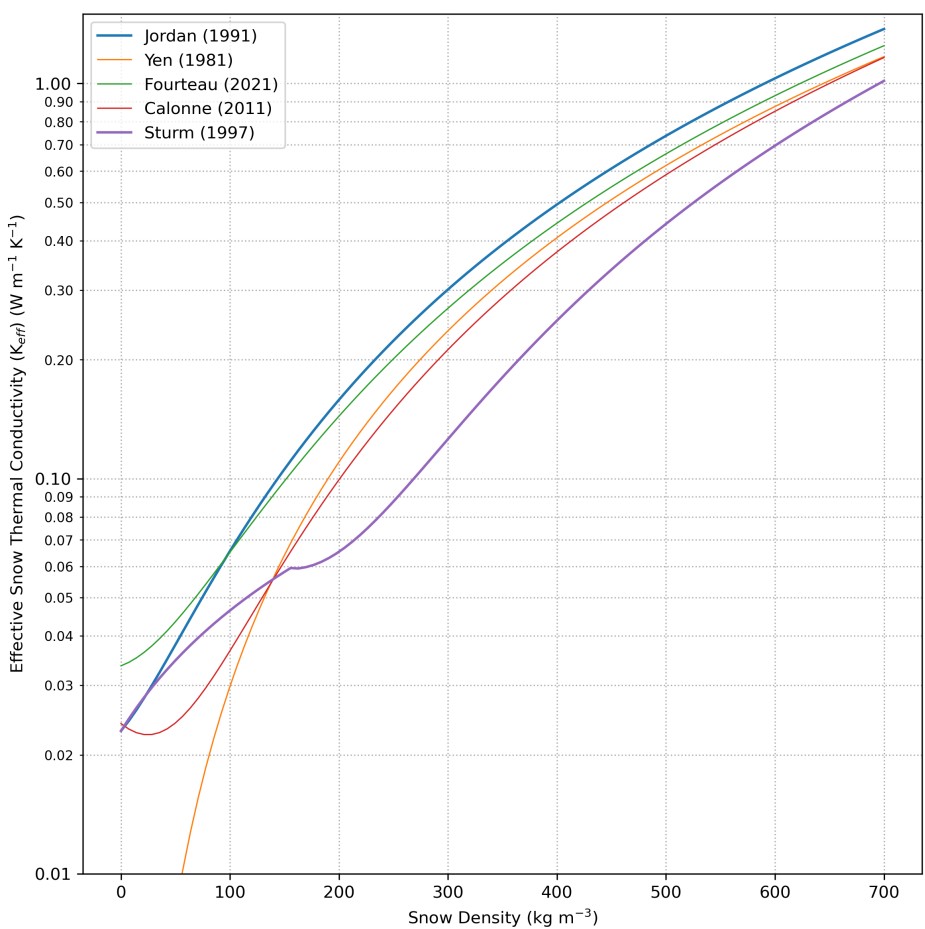

**Figure A1.** Comparison of five schemes for $K_{\text{eff}}$ from 0 to 700 $kgm^{-3}$ for snow density. Note that the y-axis is logarithmic.

Figure A1 provides a comparison of five different schemes for effective thermal conductivity ($K_{\text{eff}}$) across a range of snow densities from 0 to 700 $kgm^{-3}$. The Sturm scheme demonstrates lower $K_{\text{eff}}$ values in comparison to the other schemes, particularly within the range of snow densities encountered in permafrost regions that typically fall between 200 to 300 $kgm^{-3}$.

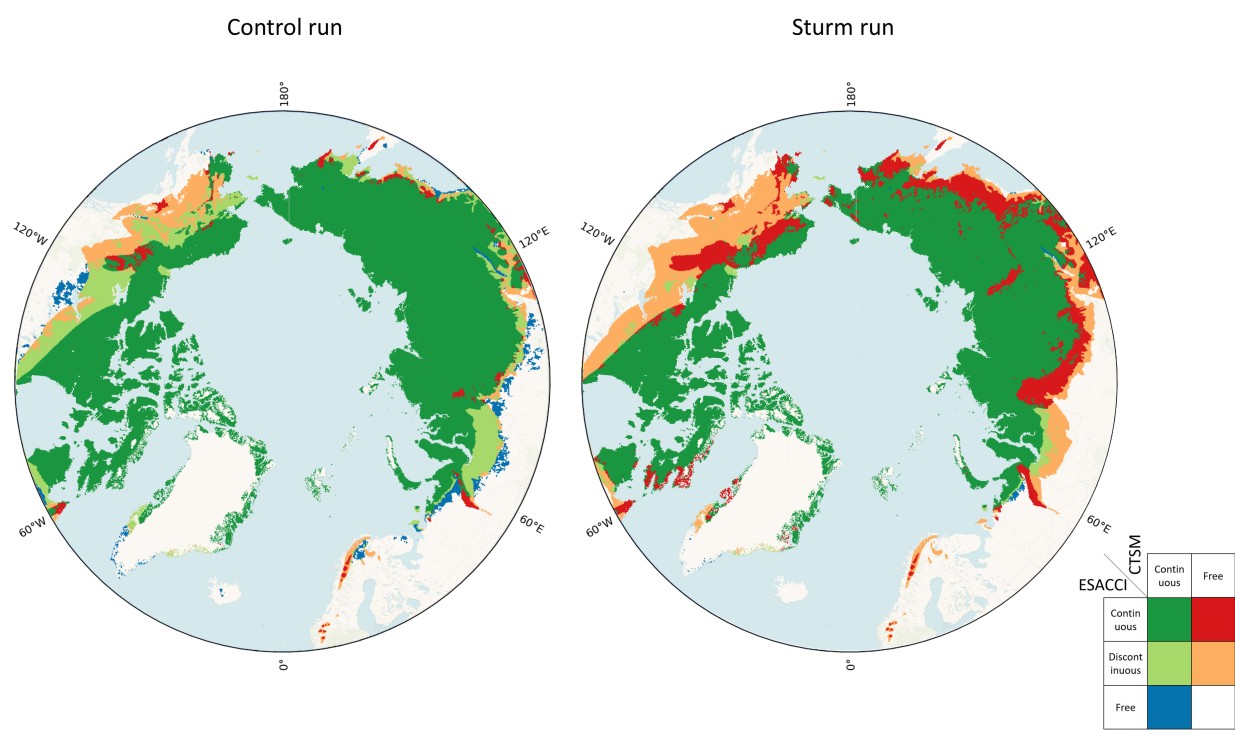

**Figure A2.** Permafrost extent area difference between CTSM control and Sturm runs (1981-1999), and the Brown et al. (2002) map.

## A2 Comparison against the permafrost extent Brown map

The IPA categorises permafrost into four distinct classes based on its areal coverage: continuous permafrost (90-100%), discontinuous permafrost (50-90%), sporadic permafrost (10-50%), and isolated permafrost (less than 10%). Similar to our comparison with ESA-CCI, we compare the continuous and discontinuous IPA categories, and assumed areas below 50% coverage to be permafrost-free to align with our binary definition of permafrost.

The permafrost extent estimated in Brown et al. (2002) surpasses that of ESA-CCI data across southern Siberia, resulting in a nearly negligible overestimation in the control run over this area (Fig. A2). However, the model fails to capture a substantial portion of discontinuous permafrost over southern Alaska.

As expected, this discrepancy leads to a more pronounced underestimation of permafrost extent in the Sturm run in many regions including Alaska, southern Canada, and southern Siberia, alongside previously mentioned areas compared to ESA-CCI products.

It is worth noting that this comparison may be less practical than with ESA-CCI products. Brown et al. (2002) data, compiled and digitised in the 1990s from historical records, represent an estimate of permafrost extent during the latter half of the

twentieth century (Burke et al., 2013). They are compared with model results covering the period 1981-1999, suggesting a potentially lower permafrost extent than in the latter half of the twentieth century.

## A3   Effective snow depth in the Sturm and control runs

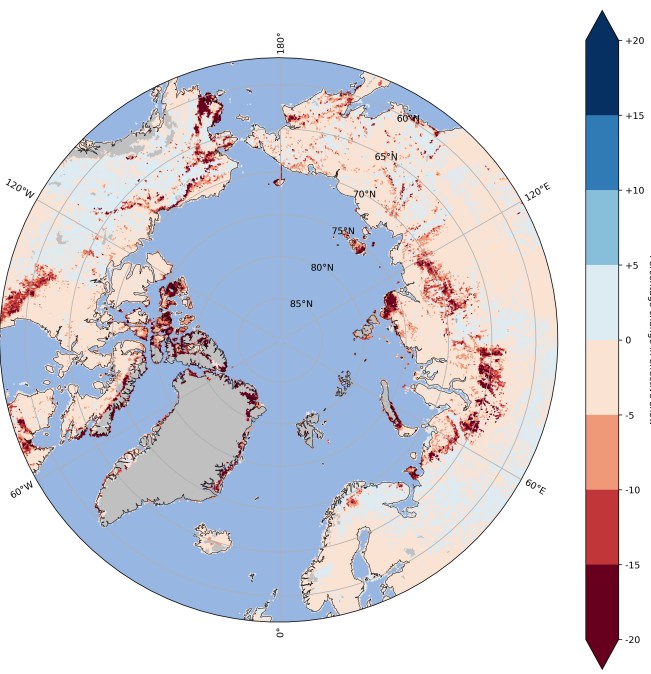

**Figure A3.** Percent change in effective snow (1980-2021 period average) between the control and Sturm runs. Darker red indicates that the Sturm run effective snow is lower than the control run. The grey mask represents glaciers.

The effective snow depth characterises the insulation provided by snow during the cold period (Burke et al., 2020). $S_{\text{depth,eff}}$ is a cumulative value where the average snow depth in each month, denoted as $S_m$ in meters, is adjusted according to its duration:

$$S_{\text{depth,eff}} = \frac{\sum_{m=1}^{M} S_m(M+1-m)}{\sum_{m=1}^{M} m} \tag{A1}$$

Snow can be present anytime from October (m = 1) to March (m = 6) with the maximum duration, $M$, being 6 months. This weighting approach favors early snowfall over late snowfall, as it contributes more to the overall insulating effect. When the effective snow depth, $S_{\text{depth,eff}}$, surpasses 0.25 meters, the insulating capacity of the snow remains relatively constant (Burke et al., 2020), and seasons with earlier snowfall typically exhibit higher $S_{\text{depth,eff}}$ than seasons with later snowfall.

Figure A3 shows the period-average percentage change in effective snow depth between the control and Sturm simulations, highlighting that there are few regions with percent changes higher than +5 or lower than -5. Percentage change is calculated as:

$$\text{Percentage change} = \left( \frac{S_{\text{depth,eff,sturm}} - S_{\text{depth,eff,control}}}{S_{\text{depth,eff,control}}} \right) \times 100 \tag{A2}$$

*Author contributions.* AD carried out the model experiments and evaluations, and wrote the original draft of the paper. AD and HM collected the different data for model evaluation. HM supervised the project. LW contributed to modifying the code in the model experiment. All authors developed the idea that lead to this paper, and were involved in reviewing and editing the paper.

*Competing interests.* The authors declare no competing interests.

*Acknowledgements.* This study was supported by the AWI INSPIRES project. AD would like to thank Evie Morin for contributing to the editing and proofreading of the manuscript. HM was supported by the European Union's Horizon2020 Program SOCIETAL CHALLENGES, grant agreement no 869471. NR and LW have been supported by the Natural Environment Research Council (Carbon Emissions under Arctic Snow - grant no. NE/W003686/1).

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
