# Peer review of "Impact of Snow Thermal Conductivity Schemes on pan-Arctic Permafrost Dynamics in CLM5.0"

_EGUsphere, 2024_

## Author Response (AR1)

**Author Response for "Impact of Snow Thermal Conductivity Schemes on pan-Arctic Permafrost Dynamics in CLM5.0" by Damseaux et al.**

We extend our gratitude to the editor and the reviewers for their thorough evaluation and insightful comments on our original manuscript. Below, we address each comment in detail. To facilitate readability, the reviewers' comments are presented in black, and our responses are in blue. References cited in this document are either found in the original manuscript bibliography or listed at the end of this document.

Reviewer 1

**General comments**

The paper of Damseaux et al. presents improved soil temperature and permafrost extent estimates using a new snow thermal conductivity scheme within CLM5.0. They highlight the importance of improved pan-arctic parametrisation in ecosystem models to achieve more realistic and robust predictions on permafrost thaw-related emissions and their impact on a regional and global scale. This study is within the scope of the journal, addresses a relevant research question, and will be of interest to the broader modelling community. I find that the article is well-written, and the results are clearly presented and discussed, however, I have some comments for the authors.

**Specific comments**

1. The authors discussed the improved model performance using the Sturm et al. (1997) instead of the Jordan (1991) scheme to calculate snow thermal conductivity. I think it would be relevant to highlight that even though the implementation of the Sturm scheme is an efficient way of addressing the insulation capacity and soil temperature bias, the effects on model outputs will need to be re-evaluated after structural developments in the snow schemes (especially if those affect snow density, which is the basis of thermal conductivity calculation).

We agree that the implementation of the Sturm et al. (1997) scheme has indeed provided an effective way to address the insulation capacity and reduce soil temperature bias in our current model framework. Indeed, we explicitly recognize in lines 238-246 of the original manuscript the trade off in improved process representation of snow densification through explicit inclusion of wind speed, which benefits dense snowpacks (particularly the creation of firn in polar environments) but degrades the simulation of tundra snowpacks where water vapour transport (not explicitly simulated) has an important influence.

Regarding the need to re-evaluate the Sturm scheme after future developments in the snow scheme, we have added this clarification in the manuscript:

"While the Sturm parameterisation offers a substantial improvement in addressing cold biases and enhancing the simulation of snow insulation in Arctic regions, it is not a panacea. Future advancements in the CLM snow scheme, particularly in the representation of snow stratigraphy and processes such as water vapor transport, will be necessary to further refine these simulations and improve model accuracy."

2. How would changes in snowpack properties e.g. snow density, liquid water content and snow depth change the preferential fit of the Sturm scheme over other thermal conductivity schemes e.g. when looking at conditions under different future climate scenarios?

Snow density: The Sturm thermal conductivity scheme is a function of snow density (Eq. 1 & Fig. A1). Increases in snow density (in CLM 5.0 using the van Kampenhout et al. 2017 formulation) will increase the thermal conductivity of the snowpack; this relationship is seen for all common snow thermal conductivity parameterisations. The reason the Sturm parameterisation is more appropriate has more to do with the unique structure of Arctic snowpacks, which seasonally develop a low density depth hoar layer under a much higher density wind slab layer. CLM5.0 cannot currently represent the processes (vertical water vapour diffusion) that create this depth hoar layer. Therefore, the thermal conductivity of the whole snowpack using the bulk density of tundra snow (i.e. the average density of the whole snow column) is better represented in CLM5.0 using the Sturm parameterisation (formulated using extensive snow measurements in Arctic tundra). Under future climate scenarios, the preferential fit of the Sturm parameterisation is unlikely to change if the van Kampenhout snow density formulation remains.

Liquid water content: A higher proportion of liquid water in the snowpack increases its bulk thermal conductivity because water is more thermally conductive than air. Consequently, if a larger percentage of water is present, more of the snowpack can refreeze, forming ice, which is the most thermally conductive of the three phases of water. While spatially discontinuous ice lenses can often form in Arctic snowpacks, they are usually 0.5 - 2 mm thick , so account for a very small percentage of the vertical snow column. Consequently, ice lenses from melt refreeze will have limited impact on thermal conductivity. When liquid water is present in snow it has a very large impact on thermal conductivity, but this is usually during very short lived (relative to the entire winter snow season) snow melt periods of relatively shallow snowpacks. Where the impact of the Sturm parameterisation is most effective is during the much longer ~6-9 months of snow-covered winter where it controls simulated soil temperatures and therefore has relevance to permafrost and carbon fluxes.

Snow depth: Based on measurements of soil and air temperature as well as snow depth, Wang et al. (2016) and Slater et al. (2017) found that there is no relationship between snow thermal conductivity and snow depth above an effective snow depth of 25 cm; i.e., effective snow thermal conductivity does not increase beyond this threshold.

3. Consider discussing the potential consequences of the significant soil temperature changes on a pan-arctic scale already in the Discussion (currently, these are brought up shortly in the Conclusion section, L339).

We have moved the text describing implications for soil temperature changes below from the Conclusion to the Discussion.

"Moving forward, it would be valuable to investigate the impacts of the sensitivity experiment proposed here within a fully-coupled ESM. Such an approach would provide insights into the complex inter-dependencies between land, snow, and the atmosphere. For instance, changes to the representation of soil temperature could have important consequences in vegetation (Jin et al., 2021), altered river flows (Rawlins and Karmalkar, 2024), permafrost-thaw-related CO2 emissions (Dutch et al., 2024), and consequently, climate feedbacks (Schädel et al., 2024)."

**Minor comments**

L115: Can you specify why you have chosen the 0.8 m SWE limit e.g. was this motivated by observations?

We reverted the changes made by van Kampenhout et al. (2017) to the 0.8 m SWE limit in order to prevent unrealistically high snow depth values in pan-Arctic non-glaciated islands. van Kampenhout et al. (2017) increased this threshold to better represent ice sheet snow conditions.It should be noted that the 0.8 m SWE threshold is significantly above the measured SWE distribution in non-glaciated Arctic regions. Data-rich studies, such as Derksen et al. (2014), show that less than 5% of the Arctic snowpack has SWE values exceeding 0.2 m, making the 0.8 m threshold a suitable choice for our study.

 We have added these clarifications to the manuscript:

"In order to prevent unrealistic high values of snow heights observed in pan-Arctic non-glaciated islands, the snow initialization protocol was recalibrated with the snow water equivalent (SWE) reverted to its original value of 0.8 m, instead of 10 m as later proposed in van Kampenhout et al. (2017)."

L142: What is the uncertainty in the permafrost classes defined by the ESA-CCI data and how is it compared to the uncertainty of your model-data deviation?

From the land model output, we cannot produce any permafrost classes: a grid cell is either classified as permafrost or no permafrost. In addition, the grid cell size of the land model is at least 100 times the grid cell size of the ESA-CCI product, so the aggregation of the ESA-CCI product to the model grid cell size blurs the borders between the permafrost zones. Accordingly, the comparison already contains technically defined uncertainties, which is acknowledged in the manuscript in the comparison presented in Figure 7.  The vast majority of grid cells in the ESA-CCI PFR product have assigned uncertainties below 50%, which makes it unlikely that these uncertainties would outweigh the uncertainties already

included in the comparison. We have added the following text to the manuscript in the discussion of the permafrost extent comparison:

"The comparison between the ESA-CCI permafrost data and our model results involves inherent uncertainties due to differences in spatial resolution. Our land model's grid cells are approximately 100 times larger than those of the ESA-CCI product, leading to blurred boundaries when aggregating the data. Although the ESA-CCI data itself has uncertainties, with most grid cells having uncertainties below 50%, these are unlikely to outweigh the uncertainties introduced by the resolution mismatch."

Figure 1. Please add the data references in the figure caption.

We have added the data references to the figure caption as requested.

"Location of the 295 borehole stations used. The size of each point represents the number of data records per station over the whole period and for all depths. The data sets are sourced from the Permafrost Laboratory website, the GTN-P database, Nordicana D, and the Roshydromet network."

Figure 2. The Celsius sign missing next to the colour bar.

We have added the Celsius sign next to the colour bar in Figure 2.

L193-: Please revise this long sentence for improved readability.

This sentence has been reconstructed into several sentences, as follows:

"The evaluation of the -1 m year-averaged soil temperature (Fig. 5) compares results from the control and Sturm runs against the ESA-CCI dataset. The Sturm run significantly reduces the cold bias observed in the control run within tundra regions, including the West Siberian Plain, Central Siberian Plateau, Yakutsk Basin, Kolyma Lowland, and northern Canada. Similar improvements were observed at soil depths of -5 m and -10 m (not shown here)."

L206: What causes the cold bias gap in the control run?

As shown in the introduction, we hypothesize that the Jordan scheme has deficiencies under Arctic snow conditions. This leads to negative biases in ground temperatures, as thermal conductivities are too high. To test this hypothesis and to potentially improve model representation of ground temperature, we exchanged the Jordan scheme with the Sturm scheme, which is presented in the sensitivity simulation called the Sturm run. We have added the following introduction to the section results that clearly relates the presentation of the results to this hypothesis:

"Our initial hypothesis suggests that the cold bias in the control run is caused by the Jordan scheme's limitations in associating snow density with thermal conductivity under Arctic conditions, leading to higher-than-expected thermal conductivities and resulting in lower ground temperatures. To rectify this cold bias, we replaced the Jordan scheme with the Sturm scheme in the Sturm run, aiming to test whether this adjustment can improve the model's representation of ground temperature."

Figure 3. I suggest adding an observed winter offset sub-figure next to the Control and Sturm run sub-figures to compare the model fit directly to observations using the two parametrisations.

Unfortunately, we did not have access to snow depth data for each station in our dataset, so we are unable to create the figure based on observations as suggested. We also reached out to the authors of Wang et al. (2016) to reproduce their figure, but they were unable to provide us with the necessary data.

Section 3.3. How did the limitation of SWE to 0.8 m affect the simulation of permafrost fringe areas e.g. in mountainous regions? (or were these areas filtered out from your spatial analysis?)

The reverted change to 0.8 m SWE primarily affects the non-glaciated pan-Arctic islands where excessive snow accumulation was observed, as discussed in https://github.com/ESCOMP/CTSM/discussions/1893. This adjustment very likely does not impact other permafrost fringe areas, such as mountainous regions in figure 7, where no relevant changes of snow depth have been observed between the default configuration and the run with reverted changes.

L248: Consider rephrasing "This observation is particularly remarkable…". As I understand this finding supports your hypothesis that the Sturm scheme provides an improved simulation of winter offset (as you mention in L249).

These sentences have been rephrased, as follows:

"The spatial distribution of the winter offset in the Sturm run better aligns with previous findings (eg., Wang et al., 2016) compared to the control run, despite the minimal difference in effective snow depth between the two runs (below +/- 5% in most regions, see Fig. A3). This supports our hypothesis that the snow insulation in the Sturm simulation is considerably increased and generally more representative of tundra snowpacks."

L362: typo: Brown et al. (data)

The typo has been fixed.

Figure A3.: Please consider changing the colour scale for increased readability (e.g. to a gradient from light to dark with increasing effective snow depth). Add "depth" and the unit (m) on the colour bar and figure caption. Alternatively, you could present a figure with the difference in effective snow depth between the Control and Sturm runs instead of the absolute values, to better visualise the spatial differences.

As rightly suggested, we have updated Figure A3 to a percentage change figure between the Control and Sturm runs.

[Figure]

Caption: "Percent change in effective snow depth (1980-2021 period average) between the control and Sturm runs. Darker red indicates that the Sturm run effective snow is lower than the control run. The grey mask represents glaciers."

The appendix chapter has also been adjusted to account for these changes.

Figure A3. caption: missing word "depth" in: "Period (1980 to 2021) average of effective snow *depth*…"

This figure caption has been updated.

Reviewer 2

The paper introduces a new snow thermal conductivity parametrization in CLM5.0, which is believed to be more adapted to Arctic snow. The default (Jordan) and new parametrization (Sturm) are compared to one another, to ESA CCI products and to in situ soil temperature data. The authors argue that the new parametrization is more representative of tundra snowpacks and is an important development in understanding changes in permafrost.

The response of permafrost and its stored carbon stocks to climate change is an important topic and a huge source of uncertainty. One of the many reasons for this uncertainty is the role of snow thermal insulation in models. As such, the topic addressed in this paper is important.

Nevertheless, there are two major issues I would like to see addressed, both of which are related to one another and to the only variable used in the Sturm parametrisation i.e. snow density:

1. The authors state that snow density in the Arctic is poorly represented in most LSMs, including in CLM. Given that their new parametrization is based on measurements, wouldn't it require snow densities simulations to be representative of Arctic snowpack in order to be effective? In lines 239-246, the authors suggest that snowpack insulation in CLM4.5 was "acceptable" due to errors in snowpack density being compensated by high thermal conductivity. This error compensation was then disrupted with the introduction of the new snow scheme. By proposing to us a snow thermal conductivity parametrisation that works with snow densities that are still not well simulated by CLM5.0, is the Sturm parametrization also not compensating for an error elsewhere? Compensating for errors in models is widespread and can be valuable, but since CLM5.0 is widely used by the scientific community, the authors need to address this clearly in the manuscript in order for future users to understand fully the limitations and compensations involved in the new parametrization.

The reviewer raises an important point about the potential for error compensation within our proposed parametrization. While it is true that the snow densities simulated by CLM5.0 (and

most LSMs) are not fully representative of Arctic snowpacks, there have been clear improvements in CLM5.0 (Lawrence et al., 2019). This limitation exists because these models do not attempt to capture the complex stratigraphy and processes within snow layers, such as vapor transport, as mentioned in the manuscript. Accurately simulating these processes would be computationally prohibitive; therefore, a simplified approach like our proposition is necessary.

In an ideal scenario where CLM had a more accurate representation of snow density profiles, the Jordan scheme might perform adequately, as it did in CLM4.5. However, we believe the Sturm scheme would likely still perform better due to its closer alignment with Arctic snow conditions. Given the current limitations of global LSMs, including CLM5.0, this hypothesis cannot be fully tested and may not be testable in the foreseeable future. Therefore, we propose using the Sturm scheme because it better aligns with the physical characteristics of Arctic snow and has demonstrably improved model performance in our simulations.

We have added the following clarification in the manuscript:

"While the Sturm parameterisation offers a substantial improvement in addressing cold biases and enhancing the simulation of snow insulation in Arctic regions, it is not a panacea. Future advancements in the CLM snow scheme, particularly in the representation of snow stratigraphy and processes such as water vapor transport, will be necessary to further refine these simulations and improve model accuracy."

2. There is no attempt to evaluate the variable (snow density) upon which the new parametrization depends at any stage in the manuscript. Comprehensive data on Arctic in situ snow density measurements are lacking, and, even if there were any, similar scale issues as highlighted by the authors in lines 276-284 with regards to soil data would likely arise. As such, a sensitivity analysis of the snow thermal conductivity parametrization to modeled snow density is essential. This analysis would help determine whether the new parametrization is robust and will remain effective even if the cold bias introduced by the new fresh snow density function is later corrected, or if it is merely a temporary fix with some underlying empirical grounding that compensates for an error elsewhere.

We acknowledge the importance of evaluating the variable upon which the new parametrization depends, particularly snow density. While comprehensive in situ data on Arctic snow density are indeed scarce, and scale issues regarding soil data are likely to arise, our study sought to evaluate the robustness of the Sturm functional relationship within the context of a climatological analysis.

By examining a large number of data points across 41 years of simulations, we have tested the robustness of the Sturm parametrization under a wide range of snow densities, snow depths, air temperatures, and soil temperatures, across a broad range of climatological

patterns. This approach aligns with our primary interest in understanding the overall climatological impacts rather than focusing on a process-level analysis, which has been emphasized in the revised manuscript:

"Our study aims to extend Dutch et al. (2022)'s assessment to evaluate the applicability of the Sturm et al. (1997) scheme in CLM5.0 across a broader regional climatological context."

We would like to highlight that the snow density improvements in CLM5.0, as mentioned in Lawrence et al. (2019), indicate that bulk snow density is now more reasonable than it was in CLM4.5, within the data limitations noted by ourselves and the reviewer. Although a detailed process-level sensitivity analysis of snow thermal conductivity parametrization to modeled snow density would indeed be valuable, it falls outside the scope of this paper, which focuses on large-scale climatological behavior.

The Sturm scheme is grounded in empirical relationships that are more representative of Arctic snow types, particularly depth hoar and wind slab formations, rather than mid-latitude snowpacks. Though this approach may compensate for some model inaccuracies, it remains physically justified. The fitted relationships between snow density and thermal conductivity, as shown by Calonne et al. (2011), demonstrate variability based on measurement types and locations. Therefore, while the Sturm scheme provides a practical solution, it is supported by solid physical reasoning.

For consideration: Although I would like the authors to consider the following questions, they may choose to ignore them if they wish, as these questions pertain to the study's legacy and broader impact. Was this study done in collaboration with the CLM5.0 team? Does this paper document the introduction of the Sturm parametrization in CLM5.0 i.e. has it already been or will soon be added as an option for users? If so, clarification on this would be useful and would assist future users as this would contribute to good research practice as part of ongoing model documentation. If the CLM5.0 team was not involved or is unaware of this study, I wonder if the authors may be willing to explain the reasons for that; I would expect that it is easier to work with the group of the model one is trying to improve rather than in isolation.

This study was conducted in close collaboration with the CTSM team. We were particularly motivated to publish this work as we believed it might address an issue with plant mortality observed in the upcoming version of CTSM, which will be used in CESM3. This issue has been discussed in detail here: https://github.com/NCAR/LMWG_dev/discussions/3.

We have shared our findings with the CTSM team, and together, we have implemented the Sturm parameterization as an option in the current master branch of the model. However, this option still requires more testing with the coupled model and is not yet present in a published CTSM/CESM version. This is why we have not shared it in the manuscript.

We hope this addition will eventually assist future users and contribute to good research practices as part of ongoing model documentation.

**Minor comments:**

- Section 2.1 includes a brief model description of the snow module, but lacks details on soil representation. While I obtained some of this information from Lawrence et al. (2019), the reviewed paper includes "permafrost" in its title, therefore I don't think it would be superfluous to include some critical information about how soil is represented e.g. how many soil layers are there in CLM5.0, and what are their thicknesses and depths? Given the different thermal properties of mineral and organic soil, concise information about their representation would also be welcome. Such information is essential for context, particularly for statements like *"the improvement is less pronounced in deeper layers, as the properties of soil increasingly dominate snow insulation properties at depth*" (L210-212).

We have added a subsubsection "soil" in the model description:

"The model soil stratigraphy includes 25 soil layers distributed geometrically, with thinner layers at shallower depths and larger layers at greater depths up to -50 meters. CLM5.0 has an increased soil layer resolution compared to CLM4.5, particularly in the upper -3 meters, to more accurately represent the Active Layer Thickness (ALT) in permafrost areas (Lawrence et al., 2019).

The heat transfer equation (Eq. 6.4 in Lawrence et al. (2018)) is numerically solved to compute soil temperatures across the 25-layer column, assuming a heat flux of zero at the bottom of the soil column. Soil temperatures are evaluated at each time step to assess phase changes in water and account for latent heat uptake and release. Hydrological calculations are conducted in the upper 20 soil layers, while the 5 bedrock layers are impermeable to water. Vertical soil moisture transport in the model is driven by the water balance equation of the whole column system, considering infiltration, surface and subsurface runoff, gradient diffusion, gravity, canopy transpiration through root extraction, and interactions with groundwater, respecting the conservation of mass. Vertical soil water flux is computed using Darcy's Law.

The model defines soil thermal and hydraulic conductivities using mineral soil parameterizations dependent on soil texture (sand, clay, and silt fractions) and organic matter density, as derived from Hugelius et al. (2013). These fractions vary across the first 10 layers but remain constant in the subsequent 15 layers."

- L180 What does "approximately" linear" mean? L240 what is "acceptable" snow insulation? These terms are rather vague. Please define.

These terms have been changed, as follows:

"Conversely, the relationship between snow depth and winter offset is close to linear across all snow depths in the control run."

"We hypothesize that the underestimation of snowpack density by CLM4.5 (Lawrence et al., 2019) combined with the high thermal conductivity scheme from Jordan (1991) artificially resulted in snow insulation that adequately represents winter soil temperatures."

- L225 "bias increase": do you mean "warm" bias, "positive" bias…?

We mean a "positive bias". We have clarified this in the manuscript.

- Lines 290-292: Doesn't snow type depend on meteorological conditions? In this case, one might argue that in a physically-based model like CLM, it may be more logical to incorporate various snow thermal conductivities based on the meteorological conditions/variables that result in different snow types rather than including different snow types explicitly. Additionally, whether different snow schemes are needed in LSMs to represent different snow types or whether improved snow physics is required is open to debate. The authors may want to nuance these statements.

Snow type is indeed influenced by meteorological factors such as precipitation, wind speed, and air temperature, but it is also shaped by soil and land cover variables, such as soil temperature, moisture, or vegetation-controlled microclimates. While incorporating snow thermal conductivities based on these meteorological and land surface conditions could enhance model results, global models like CLM must balance accuracy with computational efficiency. Including all the physical processes required to explicitly capture such details would be computationally expensive and not practical at a global scale.

We have added this clarification in the manuscript:

"The application of different empirical snow thermal conductivity schemes based on snow types (e.g., tundra or alpine) may address this challenge. However, identifying both meteorological and land surface conditions needed for accurate application of such schemes in a global model like CLM would be challenging."

- L320- As well as, perhaps, the spatial (horizontal and vertical) variability of soil properties? Referring to my first minor comment, it is important that a paper about permafrost does not forget soil properties and the potential sources of errors coming from the soil representation.

It is true that ESMs often lack detailed representation of soil properties, which can also be a significant source of error in permafrost modeling, as we discussed in lines 273-275 of the

manuscript. However, our focus in this paper was specifically the spatial and temporal variability of snow insulation; addressing soil properties was beyond the scope of our study.

We have adapted the sentence to reflect this broader context: "However, Earth System Models (ESMs) often lack sufficient detail regarding the spatial and temporal variability of snow insulation, among other factors."

Additionally, we conducted an unpublished sensitivity experiment on soil thermal properties, by altering soil texture and organic matter density default input datasets. This experiment, named the Obu run, departs from the conventional use of coarse-resolution global data in LSMs by applying a Plant Functional Type (PFT)-based approach to derive soil texture and organic matter density. The figure below compares the results from Figure 6 with those from the Obu experiment. Our findings indicate that while soil thermal properties play a role during the summer, their impact is less significant compared to the snow insulation effects discussed in this paper, particularly during winter when the control run exhibits the most bias. This is because the insulating properties of snow cover significantly dampen soil thermal dynamics in winter, as discussed in Zhu et al. (2019).

[Figure]

Caption: "Period averaged (1980-2021) of monthly soil temperature at 4 different depths (-20, -80, -160 and -320 cm) for the observations (black), control run (blue), Obu run (green), and Sturm run (red) in °C. All values come from an average of the 278 stations through the full period."

Extra references not in our manuscript:

Derksen, C., Lemmetyinen, J., Toose, P., Silis, A., Pulliainen, J., & Sturm, M. (2014). Physical properties of Arctic versus subarctic snow: Implications for high latitude passive microwave snow water equivalent retrievals. Journal of Geophysical Research Atmospheres, 119(12), 7254–7270. doi:10.1002/2013jd021264.

Zhu, D., Ciais, P., Krinner, G., Maignan, F., Jornet Puig, A., & Hugelius, G. (2019). Controls of soil organic matter on soil thermal dynamics in the northern high latitudes. Nat Commun, 10(1), 3172. doi: 10.1038/s41467-019-11103-1.

---

## Author Response (AR2)

**Second Author Response for "Impact of Snow Thermal Conductivity Schemes on pan-Arctic Permafrost Dynamics in CLM5.0" by Damseaux et al.**

Reviewer comment:

The authors have not addressed the two major issues I raised in my initial review and their rebuttal seems to rely primarily on beliefs rather than on model experiments. For instance, they state that they "believe the Sturm scheme would likely still perform better"etc, yet concede that "this hypothesis cannot be fully tested and may not be testable in the foreseeable future." In the context of a modeling paper, this rationale is insufficient.

To address this issue, I had suggested to the authors to conduct a sensitivity analysis of the parametrization to modelled snow density. However, the authors argue that such an analysis "falls outside the scope of this paper." In my view, conducting a basic sensitivity analysis e.g. by developing a toy model —particularly when data may be unavailable—falls well within the scope of model testing and evaluation.

Additionally, the authors cite the study by Calonne et al. (2011) to justify their response. However, Calonne et al. estimated snow density using tomographic imaging, which represents the state-of-the-art approach to estimate snow density. As I mentioned in my previous review, the Sturm parameterization may well be suitable when snow density is modeled with high accuracy. However, neither the revised paper nor the authors' response offer convincing evidence that the Sturm parameterization is fit-for purpose if or when there are considerable errors in modelled snow density. Unless proven otherwise, it appears, instead, that the parametrization may simply serve to compensate for errors elsewhere.

In light of these points, my perspective remains unchanged since my previous review.

Authors answer:

We appreciate the reviewer's feedback and the opportunity to clarify our position further. Our responses aim to underscore the robustness of the Sturm parameterization in representing tundra snowpacks while addressing the limitations inherent in modeling tundra snow density.

The Sturm parameterization has been demonstrated in prior studies (e.g., Sturm et al., 1997, Dutch et al., 2022) to better represent tundra snowpacks thermal properties than the Jordan parameterization. This conclusion is not unique to our work but reflects established findings in the literature. While it is true that in our study, the Sturm parameterization may partially compensate for errors in modeled snow density, this does not imply that the Jordan parameterization would outperform Sturm under conditions of an idealized tundra snowpack.

We emphasize that the critical challenge lies not in column-averaged density—which is reasonably represented in CLM5.0—but in the vertical density structure of tundra snowpacks. Current simulations, including ours, exhibit a common issue where the upper snow layers are too low in density and the basal layers too high. This structural mismatch inherently limits our ability to evaluate the "true" impact of using Sturm versus Jordan on an accurate tundra snowpack. While a sensitivity analysis could explore the relationships between snow density, thermal conductivity, and soil temperature, it cannot address the fundamental problem of structural misrepresentation.

Additionally, the new snow density formulation in CLM5.0 proposed by van Kampenhout et al. (2017) is unlikely to change significantly in the near future. This underscores the importance of using the Sturm parameterization—a data-driven, empirically validated approach tailored to tundra snow—in regions where snow has the largest impacts on permafrost dynamics. While we explicitly caution that Sturm is not universally applicable (e.g., for polar or mountain snowpacks), it remains best suited for tundra snow. Tundra regions represent the majority of global seasonal snow mass and have the most significant influence on permafrost conditions, which are critical to the Arctic climate system.

To address the author's primary concern regarding the sensitivity of the Sturm parameterization to modeled snow density, we have conducted a new sensitivity analysis. Specifically, we performed simulations using both the Sturm and Jordan thermal conductivity parameterizations, modifying snow density by scaling factors of 0.9 and 0.7 to represent lower bulk snow densities typical of tundra environments. The description, results, and discussion of this analysis have been incorporated into the different sections of the manuscript, along with a new figure illustrating the RMSE comparisons for the different configurations, and explicitly address your question regarding the performance of the Sturm parameterization under conditions of reduced snow density. The text is as follows:

Introduction section:

"Additionally, we conduct a sensitivity analysis on snow density to test the robustness of our results for potential lower bulk snow densities characteristic of tundra environments."

Methods and data section:

"To assess the sensitivity of model outputs to snow density, additional simulations were performed using both the Sturm and Jordan thermal conductivity schemes, with adjustment factors of 0.9 and 0.7 applied to the snow density parameterization to better represent the lower bulk snow densities characteristic of tundra environments. In CLM5.0, the snow density is computed as follows:

$$\rho_{sno} = af \cdot (\omega(ice) + \omega(liq)) / (frac(snow) \cdot d(z))$$

where af is the adjustment factor used in this sensitivity analysis, $\omega(ice)$ is the ice lens mass per unit area in kg/m2, $\omega(liq)$ is the liquid water mass per unit area in kg/m2, frac(snow) is the fractional snow-covered area, and d(z) is the snow layer depth in m.

The simulations were conducted exclusively for the 2006–2010 period, selected due to its robust observational data availability, to balance computational efficiency with model reliability. The four additional runs include: (1) Sturm with af= 0.9, (2) Jordan with af= 0.9, (3) Sturm with af= 0.7, and (4) Jordan with af= 0.7. These sensitivity runs were compared to baseline simulations (with af= 1.0) as part of a broader analysis of snow density impacts on model performance."

Result section:

[Figure]

**Figure 9.** Period-averaged (2006–2010) differences in monthly soil temperature RMSE (Sturm minus Jordan) across 295 stations. Each row represents a different depth (at -20, -80, -160, and -320 cm), while each column represents a different month average. Each cell represents a different scaling factor: 0.7 scaling (top), 0.9 (middle), and no scaling - default (bottom). Rectangles with positive MAD values in the Sturm run (overshoots) are marked with an asterisk (*). Darker blue indicates improved RMSE scores in Sturm relative to Jordan.

"The sensitivity analysis to snow density shows that the Sturm parameterisation regularly yields lower RMSE values compared to Jordan (blue cells in Fig. 7). This improvement is most pronounced during winter months (FMA) in deeper layers of soil. As snow density is reduced, the relative benefit of Sturm over Jordan diminishes, particularly in JFMA months at soil depths of -20 cm and -80 cm. However, the Sturm parameterisation leads to a lower soil temperature error for most months and depths. During summer months (without snow cover), the winter influence of the Sturm parametrization continues, simulating a lower temperature error than that of Jordan, particularly in deeper soil layers."

Discussion section:

"As previously stated, studies show that state-of-the-art LSMs and snowpack models, including CLM5.0, have vertical density profiles often exhibiting significant discrepancies from observed snow density, both in the top wind slab and bottom depth hoar layers of the snowpack. Such discrepancies lead to over-densification in the simulated tundra snowpack. The misrepresentation arises because the scheme does not account for temperature-gradient metamorphism, a process that creates low-density depth hoar layers in tundra snowpacks (Dutch et al., 2022). Without this mechanism, the simulated snow can only increase in density with age, leading to bulk densities that exceed observed values in these regions. Incorporating temperature-gradient metamorphism in future model developments would likely result in lower simulated snow densities, improving agreement with field observations (Brondex et al., 2023).

Our sensitivity analysis shows that the RMSE reductions achieved by the Sturm parametrization remain robust, even if future improvements are made to tundra snow densification processes that result in lower bulk densities. This improvement is most pronounced in deeper layers during winter months (FMA), when the cold wave penetrates deeply, emphasizing the relevance for permafrost modelling. This suggests that the improved performance of Sturm over Jordan does not rely on unrealistically high bulk snow density values. However, the increase in RMSE caused by the overestimation of soil temperatures in upper layers during winter months is amplified when snow density is reduced. While this highlights a limitation of the Sturm scheme in certain scenarios, the overall benefits for permafrost modelling outweigh this drawback, particularly in the context of deeper soil layers where winter thermal dynamics are critical."

---

## Author Response (AR3)

**Third Author Response for "Impact of Snow Thermal Conductivity Schemes on pan-Arctic Permafrost Dynamics in CLM5.0" by Damseaux et al.**

Editor comment:

Dear Authors,

thank you very much for providing a revised version of your manuscript. I think that the inclusion and discussion of the sensitivity analysis concerning the likely overestimation of the snow densities addresses the remaining concerns adequately. However, since reviewer #2 chose not to review the manuscript again, I would have one final request. For the sensitivity range you investigated, I would agree with your conclusion that the relative benefits of the Sturm approach outweigh the drawbacks. But I could imagine that for an even lower af, e.g. 0.5, this is no longer the case, especially when looking at the near-surface temperatures. Would it be possible for you to justify your choice of the af-range investigated, possibly providing a rough estimate of the factor by which lsms tend to overestimate the snowpack density? If that is not possible, it may be quite informative to have one additional set of simulations with a lower af, in which the annually averaged RMSE in the 0-20cm layer is higher in the Sturm runs. In this way you could define the density(-error)-range for which the Sturm scheme should be the preferred choice.

I hope that the above is not an undue request and look forward to your answer.

Sincerely,

Philipp de Vrese

Authors answer:

Dear Editor,

Thank you for your thoughtful comment. We agree that it is appropriate to justify the choice of adjustment factors in our sensitivity analysis. The selection of adjustment factors (*af*) of 0.7 and 0.9 is based on observed snow density values in Arctic tundra regions. While CLM5.0 estimates an average bulk snow density of 311 kg/m³ over our study domain (Fig. A2 below), observational studies indicate that tundra snow densities should be lower. Zhao et al. (2023) reported an average tundra bulk snow density of 225 kg/m³ based on a large dataset of Arctic-wide snow sites, which aligns well with depth hoar density measurements from multi-site (Derksen et al. 2014) and single-site studies (Woolley et al. 2024), both of which report values around 228 kg/m³. In our sensitivity analysis, an *af* of 0.7 reduces the modeled bulk snow density to 217 kg/m³, aligning well with observed densities. Meanwhile, an *af* of 0.9 results in a bulk snow density of 279 kg/m³, representing an intermediate value between the typical

LSM-simulated densities and the observed tundra densities. We will update the manuscript as follows to include this justification:

"The choice of adjustment factors is based on observed snow density values in Arctic tundra regions. CLM5.0 simulates an average bulk snow density of 311 kg/m³ over our study domain (Fig. A2), whereas observational studies indicate that tundra snow densities should be significantly lower. Zhao et al. (2023) reported an average tundra bulk snow density of 225 kg/m³ using a large dataset of Arctic-wide snow sites, while depth hoar density measurements from multi-site (Derksen et al. 2014) and single-site studies (Woolley et al. 2024) both report values around 228 kg/m³. To align model outputs with these observations, an $af$ of 0.7 was chosen to represent the lower range of observed densities, yielding a modeled bulk snow density of 217 kg/m³. Additionally, an $af$ of 0.9 was selected as an intermediate value between CLM5.0 simulated densities and the observed tundra densities."

Given this, we believe an af of 0.5 would lead to unrealistically low bulk densities, which are not representative of tundra snowpack conditions.

Appendix:

[Figure]

**Figure A2.** a) Period averaged (1980-2021) bulk snow density for the control run and b) its corresponding histogram.

"Figure A2 represents the spatial and statistical distribution of bulk snow density for the control run in our domain. The bulk snow density is calculated using the snow water equivalent (SWE) (m) and snow depth (m) through the following equation:

$$\rho_{sno} = \rho_w \frac{\text{SWE}}{\text{SD}}$$

where $\rho_w$ is the density of liquid water (1000 kg/m³). The mean density is 311 kg/m³, with an interquartile range (P25–P75) of 216 to 380 kg/m³. The histogram reveals a multimodal distribution, indicative of different snowpack types (e.g. tundra, maritime, alpine)."

Additional literature:

Derksen, C., Lemmetyinen, J., Toose, P., Silis, A., Pulliainen, J., & Sturm, M. (2014). Physical properties of Arctic versus subarctic snow: Implications for high latitude passive microwave snow water equivalent retrievals. *Journal of Geophysical Research: Atmospheres, 119(12), 7254–7270.*

Woolley, G., Rutter, N., Wake, L., Vionnet, V., Derksen, C., Essery, R., Marsh, P., Tutton, R., Walker, B., Lafaysse, M., & Pritchard, D. (2024). Multi-physics ensemble modelling of Arctic tundra snowpack properties. *The Cryosphere, 18(12), 5685–5711.*

Zhao, W., Mu, C., Wu, X., Zhong, X., Peng, X., Liu, Y., Sun, Y., Liang, B., & Zhang, T. (2023). Spatio-Temporal Characteristics and Differences in Snow Density between the Tibet Plateau and the Arctic. *Remote Sensing, 15(16), 3976.*